# AMPO: Active Multi-Preference Optimization for Self-play Preference Selection

Taneesh Gupta [* 1]   Rahul Madhavan [* 2]   Xuchao Zhang [1]   Chetan Bansal [1]   Saravan Rajmohan [1]

## Abstract

Multi-preference optimization enriches language-model alignment beyond pairwise preferences by contrasting entire sets of helpful and undesired responses, enabling richer training signals for large language models. During self-play alignment, these models often produce numerous candidate answers per query, making it computationally infeasible to include all of them in the training objective. We propose *Active Multi-Preference Optimization* (AMPO), which combines *on-policy* generation, a multi-preference *group-contrastive* loss, and *active* subset selection. Specifically, we score and embed large candidate pools of responses, then pick a small but informative subset—covering reward extremes and distinct semantic clusters—for preference optimization. The resulting contrastive training scheme identifies not only the best and worst answers but also subtle, underexplored modes crucial for robust alignment. Theoretically, we provide guarantees of expected reward maximization using our active selection method. Empirically, AMPO achieves state-of-the-art results on *AlpacaEval* with Llama 8B and Mistral 7B. We release our datasets here.

## 1. Introduction

Preference Optimization (PO) has become a standard approach for aligning large language models (LLMs) with human preferences (Christiano et al., 2017; Ouyang et al., 2022; Bai et al., 2022). Traditional alignment pipelines typically rely on pairwise or binary preference comparisons, which may not fully capture the subtleties of human judgment (Rafailov et al., 2024; Liu et al., 2024a; Korbak et al., 2023). As a remedy, there is increasing interest in *multi-preference* methods, which consider entire sets of responses

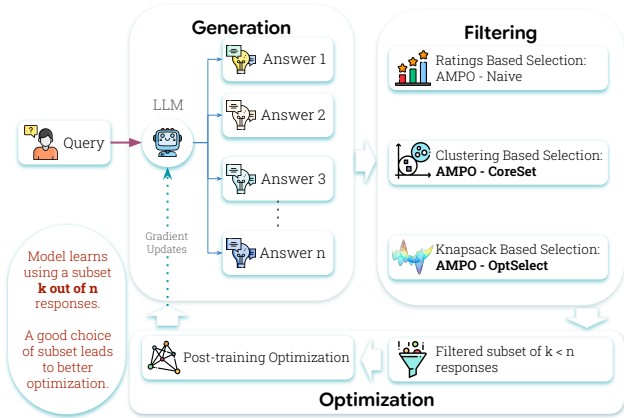

*Figure 1.* Overview of the Active Multi-Preference Optimization framework. Given a query, the LLM generates diverse responses, which are evaluated by a rater model. Selected responses with different ratings and semantics are then used to train and align the LLM through preference optimization. Active selection of the preferences to optimize over improves training dynamics.

when providing feedback (Cui et al., 2023; Chen et al., 2024a; Gupta et al., 2024b). By learning from multiple "good" and "bad" outputs simultaneously, these approaches deliver richer alignment signals. At the same time, an important trend in alignment is the shift to *on-policy* data generation, where the policy learns directly from its own distribution of outputs at each iteration (Chen et al., 2024b; Kumar et al., 2024; Wu et al., 2023; 2024). This feedback loop can accelerate convergence ensuring that the training data stays relevant to the model's behavior.

However, multi-preference alignment faces a serious bottleneck: modern LLMs can easily generate dozens of candidate responses per query, and incorporating *all* of these into a single training objective can become computationally infeasible (Askell et al., 2021). Many of these sampled responses end up being highly similar or near-duplicates, providing limited additional information for gradient updates (Long et al., 2024). Consequently, naive attempts to process all generated responses cause both memory blow-ups and diminishing returns in training (Dubey et al., 2024). Given these constraints, identifying a *small yet highly informative* subset of candidate responses is critical for effective multi-preference learning.

To understand the challenge of selecting informative responses, consider the query's answer space as a *response*

---

[*]Equal contribution [1]Microsoft [2]IISc, Bangalore. Correspondence to: Taneesh Gupta <t-taneegupta@microsoft.com>, Rahul Madhavan <mrahul@iisc.com>.

*Proceedings of the 42nd International Conference on Machine Learning*, Vancouver, Canada. PMLR 267, 2025. Copyright 2025 by the author(s).

*landscape* (See Figure 2). Each point in this landscape represents a possible response, characterized by its semantic properties (its location in the embedding space) and its quality (determined by a reward model). Furthermore, the LLM's current policy defines a probability density over this landscape. A naive approach of randomly sampling responses and treating them equally might overemphasize frequently generated areas, even if they contain only mediocre or slightly problematic answers. This risks overlooking critical feedback from less common, yet highly informative, regions—such as subtle failure points in underexplored semantic terrains, or exceptionally good responses that are rarely generated. Therefore, an ideal selection strategy must actively *explore* this landscape, identifying responses that are not just "good" or "bad" but also semantically distinct, covering reward extremes, and exposing underexplored modes that are crucial for robust alignment (Yu et al., 2024). In this paper, we show that this targeted selection can be tied to an *optimal* way of suppressing undesired modes under a mild Lipschitz assumption (see Section 7).

At its core, the problem of efficiently selecting the most impactful responses for feedback aligns with the principles of *active learning* (Cohn et al., 1996; Ceravolo et al., 2024; Xiao et al., 2023). By selecting a small yet semantically diverse subset of responses, the model effectively creates a *curriculum* for itself. Rather than passively training on random or exhaustively sampled data, an active learner *queries* the examples that yield the greatest improvement when labeled. In our context, we actively pick a handful of responses that best illustrate extreme or underexplored behaviors – whether very good, very bad, or semantically distinct (Wu et al., 2023). This helps the model quickly eliminate problematic modes while reinforcing the most desirable responses. Crucially, we remain on-policy: after each update, the newly refined policy generates a fresh batch of responses, prompting another round of active subset selection (Liu et al., 2021).

We propose **Active Multi-Preference Optimization (AMPO)**, a framework that unifies (a) on-policy data generation, (b) group-based preference learning, and (c) *active* subset selection. Specifically, we adopt a reference-free group-contrastive objective known as REFA (Gupta et al., 2024a), which jointly leverages multiple "positive" and "negative" responses in a single loss term. On top of this, we explore various active selection schemes—ranging from simplest bottom-$K$ ranking (Meng et al., 2024) to coreset-based clustering (Cohen-Addad et al., 2021; 2022; Huang et al., 2019) and a more theoretically grounded "Opt-Select" method that ties coverage to maximizing expected reward. Our contributions are: (i) a unifying algorithmic pipeline for multi-preference alignment with active selection, (ii) theoretical results demonstrating that coverage of distinct clusters à la k-medoids, can serve as an *optimal*

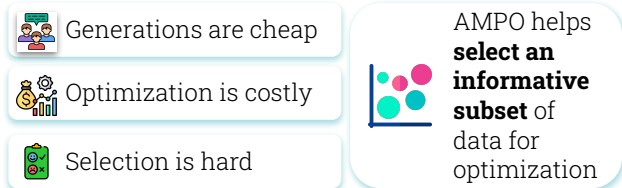

*Figure 2.* A learner can easily generate $n$ responses to a given query, but selection of a much smaller subset $k \ll n$ to train on is a hard problem. This paper addresses this problem through techniques from clustering as well as knapsack related problems.

negative-selection strategy, and (iii) empirical evaluations showing that AMPO achieves state-of-the-art results compared to strong alignment baselines like SIMPO. Altogether, our approach enables models to learn more reliably from diverse sets of model behaviors.

### 1.1. Our Contributions

- **Algorithmic Novelty:** We propose *Active Multi-Preference Optimization* (AMPO), an on-policy framework that blends group-based preference alignment with active subset selection without exhaustively training on all generated responses. This opens out avenues for research on how to select for synthetic data, as we outline in Sections 5 and 9.

- **Theoretical Insights:** Under mild Lipschitz assumptions, we show that coverage-based negative selection can systematically suppress low-reward modes and maximizes expected reward. This analysis (in Sections 6 and 7) connects our method to the weighted $K$-medoids problem, yielding performance guarantees for alignment.

- **State-of-the-Art Results:** Empirically, AMPO sets a new benchmark on *AlpacaEval* with Llama 8B, surpassing strong baselines like SIMPO by focusing on a small but strategically chosen set of responses each iteration (see Section 8).

- **Dataset Releases:** We publicly release our `AMPO-Coreset-Selection` and `AMPO-Opt-Selection` datasets on Hugging Face. These contain curated response subsets for each prompt, facilitating research on multi-preference alignment.

## 2. Related Work

Recent advances in preference optimization (Rafailov et al., 2024; Azar et al., 2023; Hong et al., 2024a) have moved beyond simple pairwise comparisons to include multiple responses per query. This shift is largely driven by datasets like UltraFeedback (Cui et al., 2023), which provide scalar

rewards for diverse candidate outputs. Within this multi-preference paradigm, methods such as InfoNCA (Chen et al., 2024a) utilize noise-contrastive objectives to align models with scalar rewards. AMPO builds upon these multi-preference approaches by employing REFA (Gupta et al., 2024a), a group-contrastive objective that contrasts sets of selected and rejected responses to emphasize multiple highly informative (positive or negative) examples.

A crucial development in LLM alignment is the adoption of *on-policy* or "self-play" data generation (Chen et al., 2024b; Wu et al., 2024). While ensuring training data relevance and accelerating convergence, this process can generate a vast number of candidate responses per query. Incorporating all these responses into the training objective becomes computationally infeasible and leads to diminishing returns due to high similarity and redundancy (Askell et al., 2021; Long et al., 2024).

To address this computational bottleneck, AMPO integrates principles from *active learning* (Cohn et al., 1996; Settles, 2009). Our active subset selection strategies draw from combinatorial optimization and clustering techniques, such as weighted $k$-medoids and coreset construction (Har-Peled & Mazumdar, 2004; Cohen-Addad et al., 2022). These methods enable AMPO to efficiently identify a small, high-impact subset of responses that effectively cover the diverse landscape of generated outputs, encompassing both reward extremes and distinct semantic regions, thereby facilitating robust and efficient alignment.

## 3. Notations and Preliminaries

On-policy alignment of LLMs with learnt preference scores often involves generating multiple candidate responses (say $N$ responses) for a given prompt. Utilizing all these $N$ candidates for training can be computationally prohibitive and may offer diminishing returns due to response similarity. Our framework, Active Multi-Preference Optimization (AMPO), addresses this by focusing on the active selection of a small, yet highly informative, subset of these responses within a pre-specified *budget* (say budget is $K$ with $K \ll N$). This section establishes the notation and foundational concepts for generating responses, evaluating them, defining selection criteria like *coverage*, and choosing subsets for efficient alignment using a group-contrastive objective.

**Queries, Policy, and Response Generation.** Let $\mathcal{D} = \{x_1, x_2, \ldots, x_M\}$ be a dataset of $M$ *queries* (or *prompts*), each from a larger space $\mathcal{X}$. We have a policy model $P_\theta(y \mid x)$, parameterized by $\theta$, which produces a distribution over possible responses $y \in \mathcal{Y}$. For each query $x_i$, we generate a pool of $N$ candidate responses $\{y_{i,1}, y_{i,2}, \ldots, y_{i,N}\}$ by sampling from $P_\theta(y \mid x_i)$ at a fixed *temperature* (e.g., Temp. = 0.8) For notational simplicity, we consider a single query $x$ and its $N$ sampled responses $\{y_1, \ldots, y_N\}$.

**Response Evaluation and Embedding.** Each response $y_j$ (for $j = 1, \ldots, N$) is assigned a scalar reward

$$r_j = \mathcal{R}(x, y_j) \in [0, 1], \tag{1}$$

where $\mathcal{R}$ is a fixed reward function. We also embed each response via $\mathbf{e}_j = \mathcal{E}(y_j) \in \mathbb{R}^d$, where $\mathcal{E}$ is an encoder capturing semantic properties. The distance between any two responses $y_j$ and $y_l$ in this embedding space is denoted $d(\mathbf{e}_j, \mathbf{e}_l)$ (e.g., Euclidean or $L2$ distance).

**Budgeted Subset Selection and Coverage.** Given the $N$ generated responses, our objective is to select a subset $\mathcal{S} \subset \{y_1, \ldots, y_N\}$ of size $K < N$, where $K$ is a pre-specified *budget* of responses to be used for training. The selection aims to maximize a utility function $\mathcal{U}$ that considers factors such as response quality (rewards), probability of generation, and embedding space coverage. Formally,

$$\mathcal{S}^* = \underset{\substack{\mathcal{S}' \subset \{y_1, \ldots, y_N\} \\ |\mathcal{S}'| = K}}{\arg \max} \; \mathcal{U}\Big(\mathcal{S}', \{r_j\}_{y_j \in \mathcal{S}'}, \{\mathbf{e}_j\}_{y_j \in \mathcal{S}'}\Big). \tag{2}$$

A key aspect of a "good" subset $\mathcal{S}^*$ is its *coverage* of the original $N$ responses. High coverage implies that for any response $y_j$ from the original $N$ candidates, its minimum distance to any response $y_l \in \mathcal{S}^*$ is small. More formally, we can define a coverage cost for a chosen subset $\mathcal{S}$ with respect to the initial $N$ responses is defined as:

$$\text{coverage\_cost}(\mathcal{S}) = \sum_{j=1}^{N} \min_{y_l \in \mathcal{S}} d(\mathbf{e}_j, \mathbf{e}_l).$$

A subset $\mathcal{S}$ provides high coverage if this sum is minimized, ensuring that the selected $K$ responses are representative of the diverse characteristics present in the initial pool of $N$. The active selection strategies discussed later (Section 5) aim to find such high-coverage subsets.

**Group-Contrastive Alignment with REFA.** Once the subset $\mathcal{S}^*$ (of size $K$) is selected, it is partitioned into a set of *accepted* responses $\mathcal{S}^+$ and a set of *rejected* responses $\mathcal{S}^-$, such that $\mathcal{S}^* = \mathcal{S}^+ \cup \mathcal{S}^-$ and $\mathcal{S}^+ \cap \mathcal{S}^- = \varnothing$. The specific criteria for this partitioning can vary (e.g., based on reward thresholds, or a one-vs-many split as in Algorithm 1). For a query $x$, we train $\theta$ using the reference-free *group-contrastive* objective REFA (Gupta et al., 2024a) by contrasting these two sets:

$$L_{\text{REFA}}(\theta) = -\log\left(\frac{\sum\limits_{y_j \in \mathcal{S}^+} \exp\Big[s_\theta\big(y_j \mid x\big)\Big]}{\sum\limits_{y_j \in (\mathcal{S}^+ \cup \mathcal{S}^-)} \exp\Big[s_\theta\big(y_j \mid x\big)\Big]}\right) \tag{3}$$

where the score for a response $y_j$, $s_\theta\big(y_j \mid x\big)$, incorporates its log-probability under the current policy $P_\theta(y_j \mid x)$ and its associated reward $r_j$. This score is given by:

$$s_\theta\big(y_j \mid x\big) = \log P_\theta\big(y_j \mid x\big) + \alpha|r_j - \bar{r}|.$$

**Algorithm 1** AMPO: One-Positive vs. $K$-Active Negatives

1: **Input:** (1) A set of $N$ responses $\{y_i\}$ sampled from $P_\theta(y \mid x)$; (2) Their rewards $\{r_i\}$, embeddings $\{\mathbf{e}_i\}$, and probabilities $\{\pi_i\}$; (3) Number of negatives $K$, initial $P_\theta$, and hyperparameter $\alpha$
2: **Output:** (i) Positive $y_+$; (ii) Negatives $\{y_j\}_{j \in S^-}$; (iii) Updated parameters $\theta$ via REFA
3: *1. Select One Positive (Highest Reward)*
4: $i_+ \leftarrow \arg\max_{i=1,\ldots,N} r_i, \quad y_+ \leftarrow y_{i_+}$
5: *2. Choose $K$ Negatives via Active Selection*
6: $\Omega \leftarrow \{1,\ldots,N\} \setminus \{i_+\}$
7: $S^- \leftarrow \text{ACTIVESELECTION}(\Omega, \{r_i\}, \{\mathbf{e}_i\}, \{\pi_i\}, K)$
8: *3. Form One-vs.-$K$ REFA Objective*
9: $\overline{r} \leftarrow \frac{r_{i_+} + \sum_{j \in S^-} r_j}{1 + K}$
10: For each $y_i$:
11: $s'_\theta(y_i) = \log P_\theta(y_i \mid x) + \alpha |r_i - \overline{r}|$
12: $L_{\text{REFA}}(\theta) = -\log\left( \frac{\exp\left[s'_\theta(y_+)\right]}{\exp\left[s'_\theta(y_+)\right] + \sum_{j \in S^-} \exp\left[s'_\theta(y_j)\right]} \right)$
13: *4. Update Model Parameters:* $\theta \leftarrow \theta - \eta \nabla_\theta L_{\text{REFA}}(\theta)$
14: **return** The chosen positive $y_+$, the negative set $\{y_j\}_{j \in S^-}$, and the updated parameters $\theta$

---

Here, $\alpha$ is a hyperparameter scaling the influence of the reward, $\log P_\theta(y_j \mid x)$ is the generation probability of response $y_j$ given query $x$, and $\overline{r} = \text{mean}_{y_j \in S}(\mathcal{R}(x, y_j))$. REFA encourages the model to increase the collective preference score of responses in $S^+$ relative to those in $S^-$. This procedure extends to any dataset $\mathcal{D}$ by summing $L_{\text{REFA}}$ across all queries. Subsequent sections detail strategies for selecting $S^*$ and partitioning it to maximize training efficiency and alignment quality.

## 4. Algorithm and Methodology

Our methodology employs a one-vs-$K$ selection scheme: one *best* response is chosen as positive, and an *active* subroutine selects $K$ negative responses from the remaining $N-1$ candidates. This active selection must balance three key objectives:

**Probability:** High-probability responses under $P_\theta(y \mid x)$ can dominate even if suboptimal by reward.

**Rewards:** Simply selecting extremes by reward misses problematic "mediocre" outputs.

**Semantics:** Diverse but undesired responses in distant embedding regions must be penalized.

While positives reinforce a single high-reward candidate, active negative selection balances probability, reward and diversity to systematically suppress problematic regions of the response space.

**Algorithm.** Formally, let $\{y_1, \ldots, y_N\}$ be the sampled responses for a single prompt $x$. Suppose we have:

**Algorithm 2** AMPO-CORESET via k-means

1: **Input:**
2: (1) $N$ responses, each with embedding $\mathbf{e}_i \in \mathbb{R}^d$ and rating $r_i$
3: (2) Desired number of negatives $K$
4:
5: **Step 1:** *Run $K$-means on embeddings*
6: Initialize $\{\mathbf{c}_1, \ldots, \mathbf{c}_K\} \subset \mathbb{R}^d$ (e.g., via $K$-means++)
7: **repeat**
8: $\quad \pi(i) = \arg\min_{1 \le j \le K} \|\mathbf{e}_i - \mathbf{c}_j\|^2, \quad i = 1, \ldots, N$
9: $\quad \mathbf{c}_j = \frac{\sum_{i:\pi(i)=j} \mathbf{e}_i}{\sum_{i:\pi(i)=j} 1}, \quad j = 1, \ldots, K$
10: **until** convergence
11:
12: **Step 2:** *In each cluster, pick the bottom-rated response*
13: For each $j \in \{1, \ldots, K\}$, define $C_j = \{i \mid \pi(i) = j\}$
14: Then $i_j^- = \arg\min_{i \in C_j} r_i, \quad j = 1, \ldots, K$
15:
16: **Step 3:** Return negatives
17: $S^- = \{i_1^-, i_2^-, \ldots, i_K^-\}$
18: **return** $S^-$ as the set of $K$ negatives

---

1. A reward function $r_i = \mathcal{R}(x, y_i) \in [0, 1]$.
2. An embedding $\mathbf{e}_i = \mathcal{E}(y_i)$.
3. A model probability $\pi_i = P_\theta(y_i \mid x)$.

Selection algorithms may be *rating-based* selection (to identify truly poor or excellent answers) with *coverage-based* selection (to explore distinct regions in the embedding space), we expose the model to both common and outlier responses. This ensures that the REFA loss provides strong gradient signals across the spectrum of answers the model is prone to generating. In Algorithm 1, ACTIVESELECTION($\cdot$) is a generic subroutine that selects a set of $K$ "high-impact" negatives. We will detail concrete implementations (e.g. bottom-$K$ by rating, clustering-based, etc.) in later sections.

## 5. Active Subset Selection Strategies

This section details two effective strategies for actively selecting $K$ negative responses within AMPO: ***AMPO-BottomK***, which selects the lowest-rated responses, and ***AMPO-Coreset***, a clustering-based method ensuring broad semantic coverage by selecting one negative per cluster. We connect AMPO-Coreset to coreset construction literature (Section E).

### 5.1. AMPO-BottomK

*AMPO-BottomK* is the most direct approach that we use for comparison: given $N$ sampled responses and their scalar ratings $\{r_i\}_{i=1}^N$, we simply pick the $K$ lowest-rated responses as negatives. This can be expressed as:

$$S^- = \text{argtopk}_i(-r_i, K), \tag{4}$$

which identifies the $K$ indices with smallest $r_i$. Although

**Algorithm 3** AMPO-OPTSELECT via Solving MIP

1: **Input:** Candidates $\{y_i\}_{i=1}^N$ with $r_i, \mathbf{e}_i$; integer $K$
2: Compute $i_{\text{top}} = \arg\max_i r_i$
3: Let $w_i = \exp(\bar{r} - r_i)$ with $\bar{r}$ as mean reward
4: Solve Problem equation 8 to get $\{x_j^*\}, \{z_{i,j}^*\}, \{y_i^*\}$
5: Let $S_{\text{neg}} = \{ j \mid x_j^* = 1 \}$ (size $K$)
6: **return** $\{i_{\text{top}}\} \cup S_{\text{neg}}$ for REFA training

**Algorithm 4** AMPO-OPTSELECT via Coordinate Descent

1: **Input:** Set $I = \{1, \ldots, N\}$, integer $K$, distances $A_{i,j}$, rewards $\{r_i\}$
2: Find $i_{\text{top}} = \arg\max_i r_i$
3: Compute $w_i = \exp(\bar{r} - r_i)$ and $d_{i,j} = A_{i,j}$
4: Initialize a random subset $S \subseteq I \setminus \{i_{\text{top}}\}$ of size $K$
5: **while** improving **do**
6:     Swap $j_{\text{out}} \in S$ with $j_{\text{in}} \notin S$ if it decreases $\sum_{i \in I} w_i \min_{j \in S} d_{i,j}$
7: **end while**
8: **return** $S_{\text{neg}} = S$ (negatives) and $i_{\text{top}}$ (positive)

conceptually simple, this method can be quite effective when the reward function reliably indicates "bad" behavior. Furthermore to break-ties, we use minimal cosine similarity with the currently selected set.

### 5.2. AMPO-Coreset (Clustering-Based Selection)

AMPO-BOTTOMK may overlook problematic modes that are slightly better than the bottom-K, but fairly important to learn on. A diversity-driven approach, which we refer to as AMPO-CORESET, explicitly seeks coverage in the embedding space by partitioning the $N$ candidate responses into $K$ clusters and then selecting the lowest-rated response within each cluster. Formally:

$$i_j^- = \arg\min_{i \in C_j} r_i, \; j = 1, \ldots, K, \; S^- = \{i_1^-, \ldots, i_K^-\}$$

where $C_j$ is the set of responses assigned to cluster $j$ by a $K$-means algorithm (Har-Peled & Mazumdar 2004; Cohen-Addad et al. 2022; see also Section E). The pseudo-code is provided in Algorithm 2.

This approach enforces that each cluster—a potential "mode" in the response space—contributes at least one negative example. Hence, AMPO-CORESET can be interpreted as selecting *representative* negatives from diverse semantic regions, ensuring that the model is penalized for a wide variety of undesired responses.

## 6. Opt-Select: Active Subset Selection by Optimizing Expected Reward

We propose *Opt-Select*, a strategy for choosing $K$ negative responses and one positive to maximize expected reward under a Lipschitz assumption. Opt-Select models the local influence of penalizing negatives, formulating an optimization problem to suppress low-reward regions while preserving high-reward modes. We present solutions via mixed-integer programming (MIP) and local search.

### 6.1. Lipschitz-Driven Objective

Let $\{y_i\}_{i=1}^n$ be candidate responses sampled on-policy, each with reward $r_i \in [0, 1]$ and embedding $\mathbf{e}_i \in \mathbb{R}^d$. Suppose that if we *completely suppress* a response $y_j$ (i.e. set its probability to zero), all answers within distance $\|\mathbf{e}_i - \mathbf{e}_j\|$ must also decrease in probability proportionally, due to a

Lipschitz constraint on the policy. Concretely, if the distance is $d_{i,j} = \|\mathbf{e}_i - \mathbf{e}_j\|$, and the model's Lipschitz constant is $L$, then the probability of $y_i$ cannot remain above $L \, d_{i,j}$ if $y_j$ is forced to probability zero.

From an *expected reward* perspective, assigning zero probability to *low-reward* responses (and their neighborhoods) improves overall alignment. To capture this rigorously, observe that the *penalty* from retaining a below-average answer $y_i$ can be weighted by:

$$w_i = \exp(\bar{r} - r_i), \tag{5}$$

where $\bar{r}$ is (for instance) the mean reward of $\{r_i\}$. Intuitively, $w_i$ is larger for lower-reward $y_i$, indicating it is more harmful to let $y_i$ and its neighborhood remain at high probability.

Next, define a distance matrix

$$A_{i,j} = \|\mathbf{e}_i - \mathbf{e}_j\|_2, \quad 1 \le i, j \le N. \tag{6}$$

Selecting a subset $S \subseteq \{1, \ldots, N\}$ of "negatives" to penalize suppresses the probability of each $i$ in proportion to $\min_{j \in S} A_{i,j}$. Consequently, a natural *cost* function measures how much "weighted distance" $y_i$ has to its closest chosen negative:

$$\text{Cost}(S) = \sum_{i=1}^N w_i \min_{j \in S} A_{i,j}. \tag{7}$$

Minimizing equation 7 yields a subset $S$ of size $K$ that "covers" or "suppresses" as many low-reward responses (large $w_i$) as possible. We then *add* one *positive* index $i_{\text{top}}$ with the highest $r_i$ to amplify a top-quality answer. This combination of *one positive* plus $K$ *negatives* provides a strong signal in the training loss.

**Interpretation and Connection to Weighted k-medoids.** If each negative $j$ "covers" responses $i$ within some radius (or cost) $A_{i,j}$, then equation 7 is analogous to a weighted $K$-*medoid* objective, where we choose $K$ items (negatives) to minimize a total weighted distance. Formally, this can be cast as a mixed-integer program (MIP) (Problem 8 below). For large $N$, local search offers an efficient approximation.

## 6.2. Mixed-Integer Programming Formulation

Define binary indicators $x_j = 1$ if we choose $y_j$ as a negative, and $z_{i,j} = 1$ if $i$ is assigned to $j$ (i.e. $\min_{j \in S} A_{i,j}$ is realized by $j$). We write:

$$\textbf{Problem } \mathcal{P} : \quad \min_{x_j \in \{0,1\}, \, z_{i,j} \in \{0,1\}, \, y_i \geq 0} \sum_{i=1}^{N} w_i \, y_i \quad (8)$$

$$\text{s.t.} \quad \sum_{j=1}^{N} x_j = K, z_{i,j} \leq x_j, \sum_{j=1}^{N} z_{i,j} = 1, \forall i,$$

$$y_i \leq A_{i,j} + M\,(1 - z_{i,j}),$$
$$y_i \geq A_{i,j} - M\,(1 - z_{i,j}), \quad \forall i,j, \quad (9)$$

where $M = \max_{i,j} A_{i,j}$. In essence, each $i$ is forced to *assign* to exactly one chosen negative $j$, making $y_i = A_{i,j}$, i.e. the distance between the answer embeddings for answer $\{i,j\}$. Minimizing $\sum_i w_i\, y_i$ (i.e. equation 7) then ensures that low-reward points ($w_i$ large) lie close to at least one penalized center.

**Algorithmic Overview.**   Solving $\mathcal{P}$ gives the $K$ negatives $S_{\text{neg}}$, while the highest-reward index $i_{\text{top}}$ is chosen as a positive. The final subset $\{i_{\text{top}}\} \cup S_{\text{neg}}$ is then passed to the REFA loss (see Section 4). Algorithm 3 outlines the procedure succinctly.

## 6.3. Local Search Approximation

For large $N$, an exact MIP can be expensive. A simpler *local search* approach initializes a random subset $\mathcal{S}$ of size $K$ and iteratively swaps elements in and out if it lowers the cost equation 7. In practice, this provides an efficient approximation, especially when $N$ or $K$ grows.

**Intuition.**   If $y_i$ is far from all penalized points $j \in S$, then it remains relatively "safe" from suppression, which is undesirable if $r_i$ is low (i.e. $w_i$ large). By systematically choosing $S$ to reduce $\sum_i w_i \min_{j \in S} d_{i,j}$, we concentrate penalization on high-impact, low-reward regions. The local search repeatedly swaps elements until no single exchange can further reduce the cost.

## 6.4. Why "Opt-Select"? A Lipschitz Argument for Expected Reward

We name the procedure "Opt-Select" because solving equation 8 (or its local search variant) directly approximates an *optimal* subset for improving the policy's expected reward. Specifically, under a Lipschitz constraint with constant $L$, assigning zero probability to each chosen negative $y_j$ implies *neighboring answers* $y_i$ at distance $d_{i,j}$ cannot exceed probability $L\,d_{i,j}$. Consequently, their contribution to the "bad behavior" portion of expected reward is bounded by

$$\exp(r_{\max} - r_i)\,(L\,d_{i,j}),$$

where $r_{\max}$ is the rating of the best-rated response. Dividing by a normalization factor (such as $\exp(r_{\max} - \overline{r})\,L$), one arrives at a cost akin to $w_i\,d_{i,j}$ with $w_i = \exp(\overline{r} - r_i)$.

*Remark* 6.1. This aligns with classical *min-knapsack* of minimizing some costs subject to some constraints, and has close alignment with the *weighted K-medoid* notions of "covering" important items at minimum cost.

# 7. Theoretical Results: Key Results

This section presents core theoretical statements underpinning AMPO's active selection. Full proofs are in Appendices C–E. We assume a budget of $K$ responses to be selected from $N$ candidates.

## 7.1. Setup and Assumptions

**(A1) $L$-Lipschitz Constraint.**   When a response $y_j$ is penalized (probability $p_j = 0$), any other response $y_i$ within embedding distance $A_{i,j}$ must satisfy $p_i \leq L\,A_{i,j}$.

**(A2) Single Positive Enforcement.**   We allow one highest-reward response $y_{i_{\text{top}}}$ to be unconstrained, i.e. $p_{i_{\text{top}}}$ is not pulled down by the negatives.

**(A3) Finite Support.**   We focus on a finite set of $N$ candidate responses $\{y_1, \ldots, y_N\}$ and their scalar rewards $\{r_i\}$, each embedded in $\mathbb{R}^d$ with distance $A_{i,j} = \|\mathbf{e}_i - \mathbf{e}_j\|$.

## 7.2. Optimal Negatives via Coverage

**Theorem 7.1** (Optimality of OPT-SELECT). *Under assumptions (A1)–(A3), let $\mathcal{S}^*$ be the set of $K$ "negative" responses that* minimizes *the coverage cost*

$$\text{Cost}(\mathcal{S}) \;=\; \sum_{i=1}^{N} \exp(\overline{r} - r_i)\,\min_{j \in \mathcal{S}} A_{i,j}, \quad (10)$$

*where $\overline{r}$ is a reference reward (e.g. average of $\{r_i\}$). Then $\mathcal{S}^*$ also* maximizes *the expected reward among all Lipschitz-compliant policies of size $K$ (with a single positive). Consequently, selecting $\mathcal{S}^*$ and allowing $p_{i_{\text{top}}} \approx 1$ is optimal.*

**Sketch of Proof.**   (See Appendix C for details.) We show a one-to-one correspondence between minimizing coverage cost $\sum_i w_i \min_{j \in \mathcal{S}} A_{i,j}$ and maximizing the feasible expected reward $\sum_i r_i p_i$ under the Lipschitz constraint. Low-reward responses with large $w_i$ must lie close to at least one negative $j \in \mathcal{S}$; else, they are not sufficiently suppressed. A mixed-integer program encodes this cost explicitly, and solving it yields the unique $\mathcal{S}^*$ that maximizes reward.

## 7.3. Local Search for Weighted $K$-Medoids

**(A4) Weighted $K$-Medoids Setup.**   We have $N$ points $\{1, \ldots, N\}$ in a metric space with distance $d(\cdot, \cdot) \geq 0$, each with weight $w_i \geq 0$. Our goal is to find a subset $\mathcal{S}$ of size $K$ to minimize $\text{Cost}\,\mathcal{S} \;=\; \sum_{i=1}^{N} w_i \min_{j \in \mathcal{S}} d(i,j)$.

| Method | Mistral-Instruct (7B) | | | | Llama-3-Instruct (8B) | | | |
| | AlpacaEval 2 | | Arena-Hard | MT-Bench | AlpacaEval 2 | | Arena-Hard | MT-Bench |
| | LC (%) | WR (%) | WR (%) | GPT-4 | LC (%) | WR (%) | WR (%) | GPT-4 |
|---|---|---|---|---|---|---|---|---|
| Base | 17.1 | 14.7 | 12.6 | 7.5 | 28.4 | 28.4 | 26.9 | 7.93 |
| RRHF[1] | 25.3 | 24.8 | 18.1 | 7.6 | 31.3 | 28.4 | 26.5 | 7.9 |
| SLiC-HF[1] | 24.1 | 24.6 | 18.9 | **7.8** | 26.9 | 27.5 | 26.2 | 8.1 |
| DPO[1] | 26.8 | 24.9 | 16.3 | 7.6 | 40.3 | 37.9 | 32.6 | 8.0 |
| IPO[1] | 20.3 | 20.3 | 16.2 | **7.8** | 35.6 | 35.6 | 30.5 | **8.3** |
| CPO[1] | 23.8 | 28.8 | 22.6 | 7.5 | 28.9 | 32.2 | 28.8 | 8.0 |
| KTO[1] | 24.5 | 23.6 | 17.9 | 7.7 | 33.1 | 31.8 | 26.4 | 8.2 |
| ORPO[1] | 24.5 | 24.9 | 20.8 | 7.7 | 28.5 | 27.4 | 25.8 | 8.0 |
| R-DPO[1] | 27.3 | 24.5 | 16.1 | 7.5 | 41.1 | 37.8 | 33.1 | 8.0 |
| SIMPO | 30.1 | 32.3 | 21.1 | 7.6 | 47.6 | 44.7 | 34.9 | 7.5 |
| AMPO-BottomK | 32.1 | 37.0 | 22.1 | 7.7 | 50.8 | 50.5 | 45.2 | 8.1 |
| AMPO-Coreset | 32.8 | 37.3 | 22.6 | **7.8** | **52.4** | **52.1** | **47.8** | 8.1 |
| AMPO-Opt-Select | **33.1** | **37.8** | **22.8** | 7.7 | 51.6 | 51.2 | 46.4 | 8.0 |

*Table 1.* Comparison of various preference optimization baselines on AlpacaEval, Arena-Hard, and MT-Bench benchmarks for Llama-3-Instruct (8B). LC-WR represents length-controlled win rate, and WR represents raw win rate. Best results are in **bold**, second-best are underlined. Our method (AMPO) achieves SOTA performance across all metrics, with different variants achieving either best or second-best results consistently.

**Theorem 7.2** (Local Search Approximation). *Suppose we apply a* 1-*swap local search algorithm to select $K$ medoids. Let $\widehat{S}$ be the resulting local optimum and let $S^*$ be the globally optimal subset. Then*

$$\text{Cost}(\widehat{S}) \leq 5 \times \text{Cost}(S^*).$$

The running time is polynomial in $N$ and $K$.

**Sketch of Proof.** (See Appendix D for a complete proof.) Assume by contradiction that $\text{Cost}(\widehat{S}) > 5\,\text{Cost}(S^*)$. We then show there exists a profitable swap (removing some $j \in \widehat{S}$ and adding $j^* \in S^*$) that strictly decreases cost, contradicting the local optimality of $\widehat{S}$.

### 7.4. Coreset Guarantee for AMPO-Coreset

**(A5) Bounded-Diameter Clusters:** For AMPO-CORESET, we assume the $N-1$ non-positive candidate responses can be grouped into $K$ semantic clusters, each with an embedding-space diameter at most $d_{max}$.

**Intuition:** AMPO-CORESET selects one lowest-rated negative from each of the $K$ semantic clusters. Under the Lipschitz constraint (A1), penalizing this single representative from a bounded-diameter cluster (A5) effectively suppresses all other semantically similar (i.e., same-cluster) responses. This ensures broad coverage across the response landscape.

**Formal Result:** (Theorem E.1, Appendix E). The induced policy's maximum expected reward is at least

$$r_{max} - L\, d_{max}, \tag{11}$$

i.e. within additive $L d_{max}$ of the unconstrained optimum given assumptions on cluster diameter ($d_{max}$) and the policy's smoothness ($L$).

## 8. Experiments

### 8.1. Experimental Setup

**Model and Training Settings:** For our experiments, we utilize a pretrained instruction-tuned model (meta-llama/MetaLlama-3-8B-Instruct), as the SFT model. These models have undergone extensive instruction tuning, making them more capable and robust compared to the SFT models used in the Base setup. However, their reinforcement learning with human feedback (RLHF) procedures remain undisclosed, making them less transparent.

To reduce distribution shift between the SFT models and the preference optimization process, we follow the approach in (Tran et al., 2023) and generate the preference dataset using the same SFT models. This ensures that our setup is more aligned with an on-policy setting. Specifically, we utilize prompts from the UltraFeedback dataset (Cui et al., 2023) and regenerate the resonses using the SFT models. For each prompt x, we produce 32 responses by sampling from the SFT model with a sampling temperature of 0.8. We then use the reward model (Skywork/Skywork-Reward-Llama-3.1-8B-v0.2) (Liu et al., 2024b) to score all the 32 responses. Then the response are selected based on the Active Subset selection strategies a.) **AMPO-Bottomk** b.) **AMPO-Coreset** c.) **AMPO-Opt-Select**

In our experiments, we observed that tuning hyperparameters is critical for optimizing the performance . Carefully selecting hyperparameter values significantly impacts the effectiveness of these methods across various datasets.We found that setting the $\beta$ (inverse temperature) parameter in the range of 5.0 to 10.0 consistently yields strong perfor-

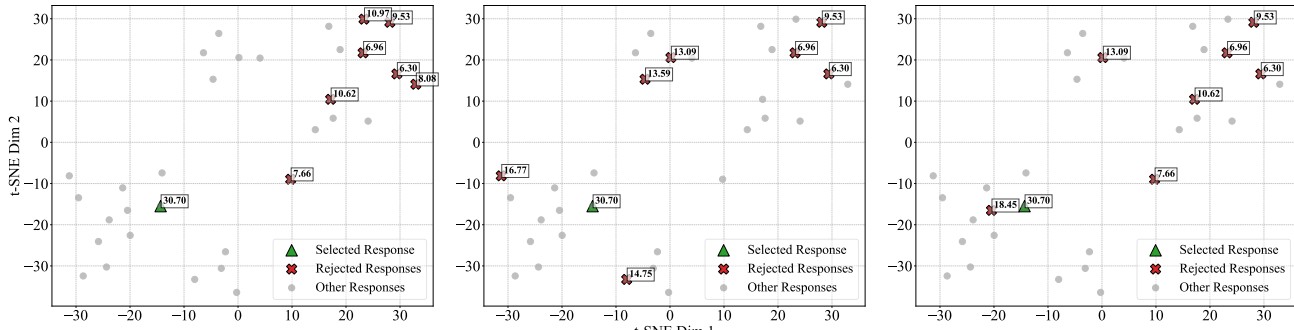

*Figure 3.* t-SNE visualization of projected high-dimensional response embeddings into a 2D space, illustrating the separation of actively selected responses. (a) AMPO-BottomK (baseline). (b) AMPO-Coreset (ours). (c) Opt-Select (ours). We see that the traditional baselines select many responses close to each other, based on their rating. This provides insufficient feedback to the LLM during preference optimization. In contrast, our methods simultaneously optimize for objectives including coverage, generation probability as well as preference rating.

mance, while tuning the $\gamma$ parameter within the range of 2 to 4 further improved performance. These observations highlight the importance of systematic hyperparameter tuning to achieve reliable outcomes across diverse datasets.

**Evaluation Benchmarks**  We evaluate our models using three widely recognized open-ended instruction-following benchmarks: MT-Bench (Zheng et al., 2023), AlpacaEval2 (Dubois et al., 2024), and Arena-Hard v0.1. These benchmarks are commonly used in the community to assess the conversational versatility of models across a diverse range of queries.

AlpacaEval 2 comprises 805 questions sourced from five datasets, while MT-Bench spans eight categories with a total of 80 questions. The recently introduced Arena-Hard builds upon MT-Bench, featuring 500 well-defined technical problem-solving queries designed to test more advanced capabilities.

We adhere to the evaluation protocols specific to each benchmark when reporting results. For AlpacaEval 2, we provide both the raw win rate (WR) and the length-controlled win rate (LC), with the latter being designed to mitigate the influence of model verbosity. For Arena-Hard, we report the win rate (WR) against a baseline model. For MT-Bench, we present the scores as evaluated by GPT-4-Preview-1106, which serve as the judge model.

### 8.2. Experimental Result

**Impact of Selection Strategies on Diversity.**  Figure 3 shows a t-SNE projection of response embeddings, highlighting how each selection method samples the answer space:

**AMPO-BottomK**: Tends to pick a tight cluster of low-rated responses, limiting coverage and redundancy in feedback.

**AMPO-Coreset**: Uses coreset-based selection to cover more diverse regions, providing coverage of examples.

**Opt-Select**: Further balances reward extremity, and embedding coverage, yielding well-separated response clusters and more effective supervision for preference alignment.

*Key Takeaway: Figure 3 demonstrates that our selection strategies significantly improve response diversity compared to traditional baselines. By actively optimizing for coverage-aware selection, our methods mitigate redundancy in selected responses, leading to better preference modeling and enhanced LLM alignment.*

**Impact of Temperature Sampling for Different Active Selection Approaches**  To analyze the impact of temperature-controlled response sampling on different active selection approaches, we conduct an ablation study by varying the sampling temperature from 0 to 1.0 in increments of 0.25 on AlpacaEval2 benchmark as demonstrated in Figure 4. We evaluate our active selection strategies observe a general trend of declining performance with increasing temperature.

*Key Takeaway: AMPO-Coreset and AMPO-Opt-Select demonstrate robustness to temperature variations, whereas LC-WR of SimPO and bottom-k selection are more sensitive.*

**Effect of gamma for Active Selection Approaches**  To investigate the sensitivity of core-set selection to different hyper-parameter settings, we conduct an ablation study on the impact of varying the gamma as shown in Figure 5. As gamma increases from 1 to 3, we observe a consistent improvement in both LC-WR and WR scores.

*Key Takeaway: This highlights the importance of tuning gamma appropriately to maximize the effectiveness of active-selection approaches.*

**Robustness to Reward Model Choice**  To assess the robustness of AMPO to the choice of reward model, we evaluate performance using two distinct reward models: Skywork-Reward-LM and GRM-Reward-LM. Table 2 presents results

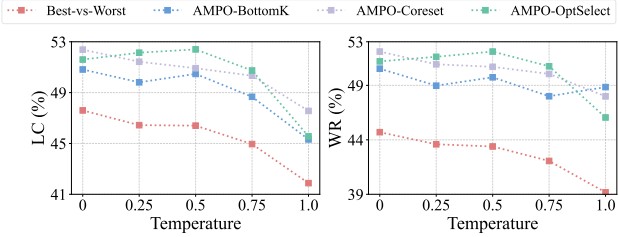

*Figure 4.* Effect of Sampling Temperature on different baselines for on the AlpacaEval 2 Benchmark: (a) Length-Controlled Win Rate (LC) and (b) Overall Win Rate (WR).

across three AMPO selection strategies—Bottom-$k$, Coreset, and Opt-Select—on AlpacaEval 2, Arena-Hard, and MT-Bench.

| Method | Reward Model | AlpacaEval 2 | |
|---|---|---|---|
| | | LC (%) | WR (%) |
| AMPO-Bottomk | Skywork-Reward-LM | 50.8 | 50.5 |
| AMPO-Coreset | Skywork-Reward-LM | **52.4** | **52.1** |
| AMPO-Opt-Select | Skywork-Reward-LM | 51.6 | 51.2 |
| AMPO-Bottomk | GRM-Reward-LM | 51.5 | 49.3 |
| AMPO-Coreset | GRM-Reward-LM | 52.5 | 49.7 |
| AMPO-Opt-Select | GRM-Reward-LM | **52.9** | **51.7** |

*Table 2.* Comparison of AMPO-baseline on AlpacaEval 2 using LLaMA-3-Instruct (8B) across different reward models

We observe that the relative ranking of methods remains largely consistent across reward models, with Opt-Select and Coreset outperforming Bottom-$k$ across metrics.

> ***Key Takeaway:*** AMPO *exhibits robust generalization across distinct reward models, indicating that its effectiveness is not tied to specific reward functions.*

**Effect of Negative Set Size ($K$) in AMPO** To examine how the number of negative comparisons affects performance, we evaluate AMPO-Opt-Select with increasing values of $K$ in the 1-vs-$K$ selection strategy—specifically, $K \in 3, 5, 7$. The results are presented in Table 3 across AlpacaEval 2, Arena-Hard, and MT-Bench.

We observe that even with a small number of negatives (e.g., $K = 3$), AMPO maintains strong performance, indicating that we identify the high-utility contrastive examples. As $K$ increases, performance improves slightly, peaking at 1-vs-7, yet the marginal gains diminish.

> ***Key Takeaway:*** AMPO *is highly effective even with a small number of negative samples, and further increases in $K$ yield diminishing returns. This shows that our method may work in resource-constrained alignment settings where generating large negative sets is costly.*

**Effect of Total Number of Responses ($N$) in AMPO-Opt-Select** We investigate the impact of varying the total number of generated responses $N$ available for selection

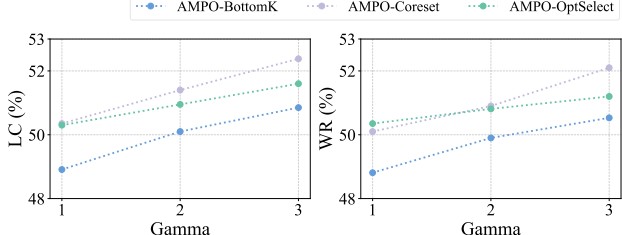

*Figure 5.* Effect of Gamma on AlpacaEval2 for Active Subset Selection Strategies.

| Method | AlpacaEval 2 | | Arena-Hard | MT-Bench |
|---|---|---|---|---|
| | LC (%) | WR (%) | WR (%) | GPT-4 |
| AMPO-Opt-Select (1vs3) | 49.6 | 48.5 | 46.1 | 8.03 |
| AMPO-Opt-Select (1vs5) | 50.3 | 49.9 | 43.9 | 7.84 |
| AMPO-Opt-Select (1vs7) | **51.6** | **51.2** | **46.4** | **8.11** |

*Table 3.* Effect of increasing the negative set size ($K$) in AMPO-Opt-Select on AlpacaEval2, Arena-Hard, and MT-Bench.

| Method | AlpacaEval 2 | | Arena-Hard | MT-Bench |
|---|---|---|---|---|
| | LC (%) | WR (%) | WR (%) | GPT-4 |
| AMPO-Opt-Select (N = 16) | 50.6 | 50.1 | 45.5 | 7.76 |
| AMPO-Opt-Select (N = 24) | 51.1 | 50.5 | 45.7 | 7.88 |
| AMPO-Opt-Select (N = 32) | **51.6** | **51.2** | **46.4** | **8.11** |

*Table 4.* Effect of increasing number of responses ($N$) for selection using AMPO-Opt-Select (1 vs 7) setting on AlpacaEval2, Arena-Hard, and MT-Bench.

in AMPO-Opt-Select under a fixed 1-vs-7 contrastive setting. Specifically, we compare performance when $N \in 16, 24, 32$, as shown in Table 4.

Our findings reveal that while increasing $N$ leads to consistent improvements across all evaluation benchmarks, the performance gains are marginal. Notably, even with $N = 16$, the results remain competitive, suggesting that AMPO-Opt-Select effectively identifies high-quality contrastive sets with limited candidate pools. Nonetheless, a larger $N$ introduces greater response diversity, which can enhance the coverage of the preference space and lead to modest performance gains—culminating at $N = 32$ with the highest scores across AlpacaEval 2, Arena-Hard, and MT-Bench.

> ***Key Takeaway:*** *Increasing the pool size of generated responses for* AMPO *improves performance, but the method remains strong even at lower $N$, demonstrating its efficiency and robustness in low-sample settings.*

## 9. Discussion & Future Work

**Iteration via Active Synthetic Data Generation.** The on-policy, coverage-focused active selection in AMPO naturally surfaces candidates for synthetic data generation. This opens avenues for future work in co-adapting the policy and reward model through this actively generated data via a robust policy-reward feedback loop.

# Impact Statement

This paper presents work whose goal is to advance the field of Machine Learning. There are many potential societal consequences of our work, none which we feel must be specifically highlighted here.

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

# SUPPLEMENTARY MATERIALS

These supplementary materials provide additional details, derivations, and experimental results for our paper. The appendix is organized as follows:

- Section A provides a more comprehensive overview of the related literature.

- Section B provides additional experiments to supplement the experiments provided in the main part of the paper.

- Section C provides theoretical analysis of the equivalence of the optimal selection integer program and the reward maximization objective.

- Section D shows a constant factor approximation for the coordinate descent algorithm in polynomial time.

- Section E provides theoretical guarantees for our k-means style coreset selection algorithm.

- Section F provides the code for computation of the optimal selection algorithm.

- Section G provides t-sne plots for the various queries highlighting the performance of our algorithms.

## A. Related Work

We start this survey with a high-level overview of the broader Reinforcement Learning from Human Feedback (RLHF) literature, then deep dive into preference optimization and multi-preference optimization, and finally discuss active learning and subset selection techniques relevant to our work.

**Preference Optimization in RLHF.** Reinforcement Learning from Human Feedback (RLHF) has emerged as a robust alignment paradigm for language models. Early methods, such as Trust Region Policy Optimization (TRPO) (Schulman et al., 2015) and Proximal Policy Optimization (PPO) (Schulman et al., 2017), extend direct RL methods by constraining policy updates for stability. PPO, in particular, has been successfully applied to RLHF, allowing LLMs to produce outputs aligned with human preferences (Ziegler et al., 2019; Ouyang et al., 2022). However, the complexity of training separate reward models and the potential instability of direct RL prompted simpler approaches.

Direct Preference Optimization (DPO) (Rafailov et al., 2024) simplifies LLM alignment by optimizing a contrastive loss directly over paired preference data, bypassing the intermediate reward modeling step. This makes DPO computationally efficient and suitable for limited preference datasets. A wide array of DPO extensions and alternative preference optimization methods have since emerged. These include variants like Identity Preference Optimization (IPO) (Azar et al., 2024), self-play preference optimization (Wu et al., 2024), preference ranking optimization (Song et al., 2024), rejection sampling optimization (Liu et al., 2023), and generalized preference optimization (Tang et al., 2024). Many of these methods also address common DPO limitations, such as the need for a fixed reference model, which adds complexity. Works like RRHF (Yuan et al., 2023) and SLiC-HF (Zhao et al., 2023) propose rank-based loss techniques. KTO (Ethayarajh et al., 2024) is a framework inspired by prospect theory that directly learns desirability, while RAFT (Dong et al., 2023) introduces a list-wise finetuning approach. Other notable methods include SPIN (Chen et al., 2024b), which treats the model as part of an adversarial game, CPO (Xu et al., 2024), which reworks the DPO objective, and ORPO (Hong et al., 2024b), which unifies SFT and preference training. SimPO (Meng et al., 2024) removes the reference model and incorporates length normalization to mitigate verbosity issues. Further variants like R-DPO (Park et al., 2024), LD-DPO (Liu et al., 2024c), sDPO (Kim et al., 2024), IRPO (Pang et al., 2024), OFS-DPO (Qi et al., 2024), and LIFT-DPO (Yuan et al., 2024) address specific challenges like length bias, reasoning chains, and training stability.

**Multi-Preference Optimization.** Traditional preference optimization methods primarily rely on pairwise comparisons. However, the advent of richer datasets, such as UltraFeedback (Cui et al., 2023), which provide multiple graded responses per query, highlights the necessity of *multi-preference optimization*. These methods move beyond simple binary preferences by leveraging all available positive and negative responses simultaneously, leading to more nuanced feedback signals (Rafailov et al., 2024; Cui et al., 2023; Chen et al., 2024a). Multi-preference objectives can reduce alignment bias and better approximate the true preference distribution by incorporating the diversity of acceptable and suboptimal responses. Examples include InfoNCA (Chen et al., 2024a), which utilizes a noise-contrastive objective based on scalar rewards. MPO (Gupta et al., 2024b), builds upon this by introducing deviation-based weighting, giving stronger influence to responses that deviate significantly (positively or negatively) from the average quality. Refa (Gupta et al., 2024a) fixes some of the common issues with MPO relating to multi-preference optimization including length bias as well as fixed reference. We build upon this framework to work on the problem of response selection in multi-preference optimization. Here, we emphasize highly informative examples while mitigating the overemphasis on less informative negative samples, a common challenge in these contrastive methods.

**On-Policy Self-Play.** A key advancement in reinforcement learning that directly impacts LLM alignment is *self-play* or on-policy data generation. In this paradigm, the model continuously updates its policy and re-generates data from its evolving distribution (Silver et al., 2016; 2017). This ensures that the training set remains aligned with the model's current behavior (Christiano et al., 2017; Wu et al., 2023; 2024), accelerating convergence and maintaining data relevance. However, this dynamic generation process can significantly inflate the number of candidate responses per query, thereby motivating the need for selective down-sampling of training examples to manage computational load.

**Active Learning for Policy Optimization.** The notion of selectively querying the most informative examples is central to *active learning* (Cohn et al., 1996; Settles, 2009), which aims to reduce labeling effort by focusing on high-utility samples. Several works incorporate active learning ideas into reinforcement learning, e.g., uncertainty sampling or diversity-based selection (Sener & Savarese, 2017; Zhang et al., 2022). In the RLHF setting, Christiano et al. (2017) highlight how strategic feedback can accelerate policy improvements, while others apply active subroutines to refine reward models (Wu et al., 2023). By picking a small yet diverse set of responses, we avoid both computational blow-ups and redundant training signals.

**Links with Classical Problems.** Our work draws heavily from classic problems in machine learning and combinatorial optimization related to selecting representative subsets. *Clustering* techniques such as $K$-means and $K$-medoids (Hartigan & Wong, 1979) are used to group points and ensure *coverage* over semantically distinct modes in the embedding space (Har-Peled & Mazumdar, 2004; Cohen-Addad et al., 2022). These methods connect to the *facility location* problem (Oh Song et al., 2017), which seeks to minimize the cost of "covering" all points with a fixed number of centers, often addressed via coreset construction (Feldman, 2020). Furthermore, when selecting a subset of size $K$ to cover or suppress "bad" outputs, the objective can be framed as a *min-knapsack* or combinatorial optimization problem (Kellerer et al., 2004a). Such formulations often involve integer programs (Chen et al., 2020), for which approximate solutions can achieve strong empirical results in high-dimensional scenarios (Cohen-Addad et al., 2022; Har-Peled & Mazumdar, 2004). Our method frames the selection of negative samples in a Lipschitz coverage sense, thereby enabling both theoretical guarantees and practical efficiency in multi-preference alignment.

Collectively, our work stands at the intersection of *multi-preference alignment* (Gupta et al., 2024a; Cui et al., 2023), *on-policy data generation* (Silver et al., 2017; Ouyang et al., 2022), and *active learning* (Cohn et al., 1996; Settles, 2009). We leverage ideas from *clustering* (k-means, k-medoids) and *combinatorial optimization* (facility location, min-knapsack) (Kellerer et al., 2004b; Cacchiani et al., 2022) to construct small yet powerful training subsets that capture both reward extremes and semantic diversity. The result is an efficient pipeline for aligning LLMs via multi-preference signals without exhaustively processing all generated responses.

# B. Additional Experiments

| Method | Reward Model | AlpacaEval 2 | | Arena-Hard | MT-Bench |
|---|---|---|---|---|---|
| | | LC (%) | WR (%) | WR (%) | GPT-4 |
| AMPO-Opt-Select-$\ell_0$ (1vsk) | Skywork-Reward-LM | 51.6 | 51.2 | **46.4** | **8.11** |
| AMPO-Opt-Select-$\ell_1$ (1vsk) | Skywork-Reward-LM | **52.16** | **51.58** | 45.4 | 8.07 |

*Table 5.* Comparison of AMPO-Opt-Select variants with and without $\ell_1$-based selection on AlpacaEval, Arena-Hard, and MT-Bench. The $\ell_1$ variant improves LC and WR on AlpacaEval, while slightly underperforming on Arena-Hard and MT-Bench. Best results are in **bold**.

**Reward Models as Classifiers vs. Regressors**    To further analyze how reward scores influence alignment, we compare two approaches to forming preference pairs and computing loss in AMPO-Opt-Select: $\ell_0$ (classification-style) and $\ell_1$ (magnitude-aware).

For both settings, we generate a fixed number of responses per prompt. The response with the highest reward is placed in the positive set $\mathcal{Y}^+$, while a contrastive negative subset is selected via Opt-Select.

- $\ell_0$ (Uniform Preference Weighting): Each preference pair contributes equally to the loss, regardless of the magnitude of reward difference. This reflects a pure classifier-style view of the reward model: only the relative ordering matters, not the exact values.

- $\ell_1$ (Reward Gap-Weighted Preference): Each preference pair is weighted by the absolute deviation of the rejected response's reward from the mean reward value, i.e., $w_i = |\text{reward}_i - \overline{\text{reward}}|$. This encourages the model to prioritize learning from examples with larger reward separation, treating them as more informative for optimization.

Table 5 presents the results. While $\ell_1$ improves AlpacaEval metrics, $\ell_0$ performs better on Arena-Hard and MT-Bench, which are known to be noisier and more ambiguous in reward calibration. These findings reinforce the hypothesis that reward magnitude may not always reflect true quality, especially when misaligned with task-specific evaluation criteria.

*Key Finding:* While weighting by reward magnitude can improve alignment on clean datasets, uniform weighting with classifier-style preferences ($\ell_0$) offers better robustness across varied and noisy evaluation settings—supporting recent trends advocating for classification-based use of reward models.

**Why 1-vs-$k$ Preference Selection is Superior to $k$-vs-$k$**    We ablate between 1-vs-$k$ and $k$-vs-$k$ preference construction strategies in AMPO, where the number of total responses is held fixed. In the 1-vs-$k$ setting, we select the single highest-scoring response as the positive and sample $k$ diverse negatives using AMPO strategies. In contrast, the $k$-vs-$k$ setup selects multiple top-scoring responses and treats them equally as positives, paired against $k$ negatives.

| Method | Reward Model | AlpacaEval 2 | | Arena-Hard | MT-Bench |
|---|---|---|---|---|---|
| | | LC (%) | WR (%) | WR (%) | GPT-4 |
| AMPO-Bottomk (1vs7) | Skywork-Reward-LM | **50.8** | 50.5 | **45.2** | **8.11** |
| AMPO-Bottomk (4vs4) | Skywork-Reward-LM | 45.44 | **51.25** | 42.2 | 7.77 |
| AMPO-Coreset (1vs7) | Skywork-Reward-LM | **52.4** | **52.1** | **47.8** | **8.12** |
| AMPO-Coreset (4vs4) | Skywork-Reward-LM | 46.61 | 51.4 | 46.3 | 7.67 |
| AMPO-Opt-Select (1vs7) | Skywork-Reward-LM | **51.6** | 51.2 | **46.4** | **8.11** |
| AMPO-Opt-Select (4vs4) | Skywork-Reward-LM | 47.16 | **52.5** | 44.9 | 7.72 |

*Table 6.* Dynamic AMPO-Based Top/Bottom Response Selection Across Evaluation Benchmarks for Llama-3-Instruct (8B)

**Theoretical Motivation** If the goal is to maximize expected reward, the optimal strategy—when only the sampling probabilities over responses can be controlled—is to assign the highest probability to the response with the maximum reward score. This ensures reward-weighted sampling favors the best response. Including multiple responses in the positive set can dilute this probability mass and introduce ambiguity, especially when the difference between top-ranked responses is small or noisy.

This analysis is formalized in Section B.1 of the paper, where we show that concentrating probability mass on the single best response, while distributing mass away from contrastive negatives, is provably optimal in expectation under a fixed response budget.

| Method | Reward Model | AlpacaEval 2 | | Arena-Hard | MT-Bench |
|---|---|---|---|---|---|
| | | LC (%) | WR (%) | WR (%) | GPT-4 |
| AMPO-Opt-Select (1vs3) | Skywork-Reward-LM | 49.6 | 48.5 | 46.1 | 8.03 |
| AMPO-Opt-Select (2vs2) | Skywork-Reward-LM | 48.64 | 47.85 | 42.1 | 7.87 |

*Table 7.* Dynamic AMPO-Based Top/Bottom Response Selection with Skywork-Reward-LM Across Evaluation Benchmarks for Llama-3-Instruct (8B).

**Empirical Evidence** We validate this hypothesis with empirical results presented in Table 6 and Table 7. Across Bottom-$k$, Coreset, and Opt-Select variants, the 1-vs-7 configuration consistently outperforms 4-vs-4, particularly in terms of LC win rate (AlpacaEval2), Arena-Hard, and MT-Bench. Similarly, 1-vs-3 outperforms 2-vs-2.

This suggests that: Selecting a single clear positive introduces less ambiguity. Including multiple positives can inject noise if some "positive" responses are marginal or inconsistent. A broader negative set ($k$ larger) allows for better contrast and generalization.

*Key Finding:* The 1-vs-$k$ preference setup is theoretically optimal for maximizing expected reward and empirically leads to better performance. This supports our design choice of using a single, high-confidence positive response when constructing preference data for alignment.

## C. Extended Theoretical Analysis of OPT-SELECT

In this appendix, we present a more detailed theoretical treatment of AMPO-OPTSELECT. We restate the core problem setup and assumptions, then provide rigorous proofs of our main results. Our exposition here augments the concise version from the main text.

### C.1. Problem Setup

Consider a single prompt (query) $x$ for which we have sampled $N$ candidate responses $\{y_1, y_2, \ldots, y_N\}$. Each response $y_i$ has:

- A scalar reward $r_i \in [0, 1]$.

- An embedding $\mathbf{e}_i \in \mathbb{R}^d$.

We define the distance between two responses $y_i$ and $y_j$ by

$$A_{i,j} \;=\; \|\mathbf{e}_i - \mathbf{e}_j\|. \tag{12}$$

Throughout we rescale the embedding so that $\max_{i,j} A_{i,j} = 1$; the Lipschitz constant $L \in (0, 1]$ then compares quantities of the same scale.

We wish to learn a *policy* $\{p_i\}$, where $p_i \geq 0$ and $\sum_{i=1}^N p_i = 1$. The policy's *expected reward* is

$$\mathrm{ER}(p) \;=\; \sum_{i=1}^N r_i\, p_i. \tag{13}$$

**Positive and Negative Responses.** We designate exactly one response, denoted $y_{i_{\mathrm{top}}}$, as a *positive* (the highest-reward candidate). All other responses are potential "negatives." Concretely:

- We fix one index $i_{\mathrm{top}}$ with $i_{\mathrm{top}} \;=\; \arg\max_{i \in \{1,\ldots,N\}} r_i$.

- We choose a subset $\mathcal{S} \subseteq \{1, \ldots, N\} \setminus \{i_{\mathrm{top}}\}$ of size $K$, whose elements are forced to have $p_j = 0$. (These are the "negatives.")

**Tie–breaking.** If several responses attain the maximal reward we keep the one with the smallest index; thus $i_{\text{top}}$ is unique.

### C.1.1. LIPSCHITZ SUPPRESSION CONSTRAINT

We assume a mild Lipschitz-like rule:

(A1) $L$-**Lipschitz Constraint.** If $p_j = 0$ for some $j \in \mathcal{S}$, then for every response $y_i$, we must have

$$p_i \leq L A_{i,j} = L \|\mathbf{e}_i - \mathbf{e}_j\|. \tag{14}$$

The effect is that whenever we force a particular negative $j$ to have $p_j = 0$, any response $i$ near $j$ in embedding space also gets *pushed down*, since $p_i \leq L A_{i,j}$. By selecting a set of $K$ negatives covering many "bad" or low-reward regions, we curb the policy's probability of generating undesirable responses.

**Goal.** Define the feasible set of distributions:

$$\mathcal{F}(\mathcal{S}) = \left\{ \{p_i\} \colon p_j = 0 \,\forall j \in \mathcal{S}, \; p_i \leq L \min_{j \in \mathcal{S}} A_{i,j} \,\forall i \notin \{i_{\text{top}}\} \cup \mathcal{S} \right\}. \tag{15}$$

**Feasibility condition.** For a given $\mathcal{S}$ the constraint set $\mathcal{F}(\mathcal{S})$ is non-empty iff

$$\sum_{i \notin \mathcal{S} \cup \{i_{\text{top}}\}} \min_{j \in \mathcal{S}} A_{i,j} \leq 1/L.$$

Hence we assume $K$ and $L$ are chosen so that the above inequality holds for at least one subset of size $K$.

We then have a two-level problem:

$$\max_{\substack{\mathcal{S} \subseteq \{1,\ldots,N\}\setminus\{i_{\text{top}}\} \\ |\mathcal{S}|=K}} \quad \max_{\substack{\{p_i\}\in\mathcal{F}(\mathcal{S}) \\ \sum_i p_i=1, \, p_i \geq 0}} \quad \sum_{i=1}^{N} r_i \, p_i,$$

$$\text{subject to} \quad p_{i_{\text{top}}} \text{ is unconstrained (no Lipschitz bound).} \tag{16}$$

We seek $\mathcal{S}$ that *maximizes* the best possible Lipschitz-compliant expected reward.

### C.2. Coverage View and the MIP Formulation

**Coverage Cost.** To highlight the crucial role of "covering" low-reward responses, define a weight

$$w_i := r_{\max} - r_i \tag{17}$$

where $r_{\max} = \max_j r_j$. Then a natural *coverage* cost is

$$\text{Cost}(\mathcal{S}) = \sum_{i=1}^{N} w_i \min_{j \in \mathcal{S}} A_{i,j}. \tag{18}$$

A small $\min_{j \in \mathcal{S}} A_{i,j}$ means response $i$ is "close" to at least one negative center $j$. If $r_i$ is low, then $w_i$ is large, so we put higher penalty on leaving $i$ uncovered. Minimizing $\text{Cost}(\mathcal{S})$ ensures that *important* (low-reward) responses are forced near penalized centers, thus *suppressing* them in the policy distribution.

**MIP $\mathcal{P}$ for Coverage Minimization.** We can write a mixed-integer program:

$$\textbf{Problem } \mathcal{P}: \min_{\substack{x_j \in \{0,1\} \\ z_{i,j} \in \{0,1\} \\ y_i \geq 0}} \sum_{i=1}^{N} w_i \, y_i,$$

$$\text{subject to } \begin{cases} \sum_{j=1}^{N} x_j = K, \\ z_{i,j} \leq x_j, \quad \sum_{j=1}^{N} z_{i,j} = 1, \quad \forall i, \\ y_i \leq A_{i,j} + M \, (1 - z_{i,j}), \\ y_i \geq A_{i,j} - M \, (1 - z_{i,j}), \quad \forall i, j, \end{cases} \tag{19}$$

where $M = \max_{i,j} A_{i,j}$. Intuitively, each $x_j$ indicates if $j$ is chosen as a negative; each $z_{i,j}$ indicates whether $i$ is "assigned" to $j$. At optimality, $y_i = \min_{j \in \mathcal{S}} A_{i,j}$, so the objective $\sum_i w_i \, y_i$ is precisely $\text{Cost}(\mathcal{S})$. Hence solving $\mathcal{P}$ yields $\mathcal{S}^*$ that *minimizes* coverage cost equation 18.

**Lemma C.1** (Coverage cost controls negative reward). *Under (A1)–(A3), suppose $S$ of size $K$ satisfies the feasibility condition, i.e. there exists $\{p_i\} \in \mathcal{F}(S)$ with $\sum_i p_i = 1$. Then for every normalized feasible $\{p_i\}$ (i.e. $\forall \{p_i\} \in \mathcal{F}(S)$), we have:*

$$\sum_{i=1}^{N} (r_{\max} - r_i) \, p_i \; \leq \; L \sum_{i=1}^{N} (r_{\max} - r_i) \min_{j \in S} A_{i,j} \; = \; L \, \text{Cost}(S),$$

*Consequently,*

$$\max_{\{p_i\} \in \mathcal{F}(S)} \sum_i r_i \, p_i = r_{\max} - \min_{\{p_i\} \in \mathcal{F}(S)} \sum_i (r_{\max} - r_i) p_i$$

*is* maximised *exactly when* $\text{Cost}(S)$ *is minimised.*

*Furthermore, this bound is tight: one can set*

$$p_i = L \min_{j \in S} A_{i,j} \quad (i \neq i_{top}), \qquad p_{i_{top}} = 1 - L \sum_{i \neq i_{top}} \min_{j \in S} A_{i,j},$$

*which is feasible by assumption, and gives $\sum_i (r_{\max} - r_i) p_i = L \, \text{Cost}(S)$, so $\max \sum_i r_i p_i = r_{\max} - L \, \text{Cost}(S)$.*

*Proof.* By (A1), any $i \notin S \cup \{i_{top}\}$ satisfies $p_i \leq L \min_{j \in S} A_{i,j}$, hence $(r_{\max} - r_i) p_i \leq L \, (r_{\max} - r_i) \min_{j \in S} A_{i,j}$. Summing over $i$ yields the claimed bound, and the equivalence between minimising $\text{Cost}(S)$ and maximising $\sum_i r_i p_i$ follows by writing

$$\sum_i r_i p_i = r_{\max} \underbrace{\sum_i p_i}_{=1} - \sum_i (r_{\max} - r_i) p_i = r_{\max} - \sum_i (r_{\max} - r_i) p_i,$$

and observing the inequality becomes an equality for the choice above. $\square$

### C.3. Main Theorem: Optimality of $\mathcal{P}$ for Lipschitz Alignment

**Theorem C.2** (Optimal Negative Set via $\mathcal{P}$). *Let $\mathcal{S}^*$ be the solution to the MIP $\mathcal{P}$ in equation 19, i.e. it* minimizes *$\text{Cost}(\mathcal{S})$. Then $\mathcal{S}^*$ also* maximizes *the objective equation 16. Consequently, picking $\mathcal{S}^*$ and allowing any distribution on $i_{\text{top}} \approx \arg\max_i r_i$ yields the* optimal *Lipschitz-compliant policy.*

*Proof.* By construction, solving $\mathcal{P}$ returns $\mathcal{S}^*$ with $\text{Cost}(\mathcal{S}^*) = \min_{|\mathcal{S}|=K} \text{Cost}(\mathcal{S})$. By Lemma C.1, minimising $\text{Cost}(S)$ indeed maximises the feasible expected reward, so such an $\mathcal{S}^*$ simultaneously *maximizes* the best possible feasible expected reward. Hence $\mathcal{S}^*$ is precisely the negative set that achieves the maximum of equation 16.

$\square$

**Interpretation.**    Under a mild Lipschitz assumption in embedding space, penalizing (assigning zero probability to) a small set $\mathcal{S}$ *and* forcing all items near $\mathcal{S}$ to have small probability is equivalent to a *coverage* problem. Solving (or approximating) $\mathcal{P}$ selects negatives that push down low-reward modes as effectively as possible.

### C.4. Discussion and Practical Implementation

OPT-SELECT thus emerges from optimizing coverage:

1. **Solve or approximate** the MIP $\mathcal{P}$ to find the best subset $\mathcal{S} \subseteq \{1, \ldots, N\} \setminus \{i_{\text{top}}\}$.

2. **Force** $p_j = 0$ for each $j \in \mathcal{S}$; **retain** $i_{\text{top}}$ with full probability ($p_{i_{\text{top}}} \approx 1$), subject to normalizing the distribution.

In practice, local search or approximate clustering-based approaches (e.g. Weighted $K$-Medoids) can find good solutions without exhaustively solving $\mathcal{P}$. The method ensures that near any chosen negative $j$, all semantically similar responses $i$ have bounded probability $p_i \leq L\, A_{i,j}$. Consequently, OPT-SELECT *simultaneously* covers and suppresses undesired modes while preserving at least one high-reward response unpenalized.

**Additional Remarks.**

- The single-positive assumption reflects a practical design where one high-reward response is explicitly promoted. This can be extended to multiple positives, e.g. top $K^+$ responses each unconstrained.

- For large $N$, the exact MIP solution may be expensive; local search (see Appendix D) still achieves a constant-factor approximation.

- The embedding-based Lipschitz constant $L$ is rarely known exactly; however, the coverage perspective remains valid for "sufficiently smooth" reward behaviors in the embedding space.

Overall, these results solidify OPT-SELECT as a principled framework for negative selection under Lipschitz-based alignment objectives.

# D. Local Search Guarantees for Weighted $K$-Medoids and Lipschitz-Reward Approximation

In this appendix, we show in Theorem D.1 that a standard *local search* algorithm for *Weighted $K$-Medoids* achieves a constant-factor approximation in polynomial time.

## D.1. Weighted $K$-Medoids Setup

We are given:

- A set of $N$ points, each indexed by $i \in \{1, \ldots, N\}$.

- A distance function $d(i, j) \geq 0$, which forms a metric: $d(i, j) \leq d(i, k) + d(k, j)$, $d(i, i) = 0$, $d(i, j) = d(j, i)$.

- A nonnegative *weight* $w_i$ for each point $i$.

- A budget $K$, $1 \leq K \leq N$.

We wish to pick a subset $\mathcal{S} \subseteq \{1, \ldots, N\}$ of *medoids* (centers) with size $|\mathcal{S}| = K$ that minimizes the objective

$$\text{Cost}(\mathcal{S}) \;=\; \sum_{i=1}^{N} w_i \cdot \min_{j \in \mathcal{S}} d(i, j). \tag{20}$$

We call this the **Weighted $K$-Medoids** problem. Note that **medoids** must come from among the data points, as opposed to $K$-median or $K$-means where centers can be arbitrary points in the metric or vector space. Our Algorithm 3 reduces to exactly this problem.

## D.2. Coordinate Descent Algorithm via Local Search

Our approach to the NP-hardness of Algorithm 3 was to recast it as a simpler coordinate descent algorithm in Algorithm 4, wherein we do a local search at every point towards achieving the optimal solution. Let $\text{COST}(\mathcal{S})$ be as in equation 20.

1. **Initialize:** pick any subset $\mathcal{S} \subseteq \{1, \ldots, N\}$ of size $K$ (e.g. random or greedy).

2. **Repeat**: Try all possible single *swaps* of the form

$$\mathcal{S}' \;=\; \big(\mathcal{S} \setminus \{j\}\big) \cup \{j'\},$$

   where $j \in \mathcal{S}$ and $j' \notin \mathcal{S}$.

3. **If any swap improves cost**: i.e. $\text{Cost}(\mathcal{S}') < \text{Cost}(\mathcal{S})$, then set $\mathcal{S} \leftarrow \mathcal{S}'$ and continue.

4. **Else terminate**: no single swap can further reduce cost.

When the algorithm stops, we say $\mathcal{S}$ is a *local optimum under 1-swaps*.

## D.3. Constant-Factor Approximation in Polynomial Time

We now present and prove a result: such local search yields a constant-factor approximation. Below, we prove a version with a *factor 5* guarantee for Weighted $K$-Medoids. Tighter analyses can improve constants, but 5 is a commonly cited bound for this simple variant.

**Theorem D.1** (Local Search for Weighted $K$-Medoids). *Let $\mathcal{S}^*$ be an **optimal** subset of medoids of size $K$. Let $\widehat{\mathcal{S}}$ be any **local optimum** obtained by the above 1-swap local search. Then*

$$\text{Cost}\big(\widehat{\mathcal{S}}\big) \;\leq\; 5 \times \text{Cost}\big(\mathcal{S}^*\big). \tag{21}$$

*Moreover, the procedure runs in polynomial time (at most $\left(\binom{N}{K}\right)$) "worse-case" swaps in principle, but in practice each improving swap decreases cost by a non-negligible amount, thus bounding the iteration count).*

*Remark* D.2. We follow the result from Arya et al. (2001) who define the *locality gap* of the single-swap local-search procedure as the worst-case ratio between the cost of any local optimum and the global optimum. They prove that for the metric K-median problem, this gap is exactly 5. More precisely, permitting only one swap per step guarantees

$$\text{Cost}(\widehat{S}) \; \leq \; 5\,\text{Cost}(S^*) \tag{22}$$

for every local optimum $\widehat{S}$ and global optimum $S^*$

**Sketch of Arya et al. (2001)'s Analysis.** They partition the data according to the Voronoi cells of the global optimum, then show via a "coupling" argument (together with repeated triangle-inequality bounds) that whenever a local swap cannot improve the solution, the total service cost from each cell is bounded by five times its optimal cost.

*Proof.* **Notation.**

- Let $\widehat{S}$ be the final local optimum of size $K$.

- Let $S^*$ be an optimal set of size $K$.

- For each point $i$, define
$$r_i \; = \; d(i, \widehat{S}) \; = \; \min_{j \in \widehat{S}} d(i,j), \quad r_i^* \; = \; \min_{j \in S^*} d(i,j).$$
Thus $\text{Cost}(\widehat{S}) = \sum_i w_i\, r_i$ and $\text{Cost}(S^*) = \sum_i w_i\, r_i^*$.

- Let $c(S) = \sum_i w_i\, d(i, S)$ as shorthand for $\text{Cost}(S)$.

**Step 1: Construct a "Combined" Set.** Consider

$$S^\dagger \; = \; \widehat{S} \cup S^*.$$

We have $|S^\dagger| \leq 2K$. Let $c(S^\dagger) = \sum_i w_i\, d(i, S^\dagger)$.

Observe that

$$d(i, S^\dagger) \; = \; \min\{d(i, \widehat{S}), d(i, S^*)\} \; = \; \min\{r_i,\, r_i^*\}.$$

Hence

$$c(S^\dagger) \; = \; \sum_{i=1}^{N} w_i\, \min\{r_i,\, r_i^*\}.$$

We will relate $c(S^\dagger)$ to $c(\widehat{S})$ and $c(S^*)$.

**Step 2: Partition Points According to $S^*$.** For each $j^* \in S^*$, define the cluster

$$C(j^*) \; = \; \{i \mid j^* = \arg\min_{j' \in S^*} d(i, j')\}.$$

Hence $\{\, C(j^*) \,:\, j^* \in S^* \}$ is a partition of $\{1, \ldots, N\}$. We now group the cost contributions by these clusters.

**Goal: Existence of a Good Swap.** We will *assume $c(\widehat{S}) > 5\,c(S^*)$* and derive a contradiction by producing a *profitable swap* that local search should have found.

Specifically, we show that there must be a center $j^* \in S^*$ whose cluster $C(j^*)$ is "costly enough" under $\widehat{S}$, so that swapping out some center $j \in \widehat{S}$ for $j^*$ significantly reduces cost. But since $\widehat{S}$ was a local optimum, no such profitable swap could exist. This contradiction implies $c(\widehat{S}) \leq 5\,c(S^*)$.

**Step 3: Detailed Bounding.**

We have

$$c(S^\dagger) = \sum_{i=1}^{N} w_i\, \min\{r_i, r_i^*\} \; \leq \; \sum_{i=1}^{N} w_i\, r_i^* = c(S^*).$$

Similarly,

$$c(\mathcal{S}^{\dagger}) \;\leq\; \sum_{i=1}^{N} w_i \, r_i = c(\widehat{\mathcal{S}}).$$

Hence $c(\mathcal{S}^{\dagger}) \leq \min\{c(\widehat{\mathcal{S}}),\, c(\mathcal{S}^*)\}$. Now define

$$D \;=\; \sum_{i=1}^{N} w_i \left[ r_i - \min\{ r_i,\, r_i^* \} \right] \;=\; \sum_{i=1}^{N} w_i \left( r_i - r_i^* \right)_+,$$

where $(x)_+ = \max\{x, 0\}$. By rearranging,

$$\sum_{i=1}^{N} w_i \, r_i \;-\; \sum_{i=1}^{N} w_i \, \min\{ r_i,\, r_i^* \} \;=\; D.$$

Thus

$$c(\widehat{\mathcal{S}}) - c(\mathcal{S}^{\dagger}) \;=\; D \;\geq\; c(\widehat{\mathcal{S}}) - c(\mathcal{S}^*).$$

So

$$D \;\geq\; c(\widehat{\mathcal{S}}) \;-\; c(\mathcal{S}^*).$$

Under the assumption $c(\widehat{\mathcal{S}}) > 5\, c(\mathcal{S}^*)$, we get

$$D \;>\; 4\, c(\mathcal{S}^*). \tag{*}$$

**Step 4: Find a Center $j^*$ with Large $D$ Contribution.** We now "distribute" $D$ over clusters $C(j^*)$. Let

$$D_{j^*} \;=\; \sum_{i \in C(j^*)} w_i \left( r_i - r_i^* \right)_+.$$

Then $D = \sum\limits_{j^* \in \mathcal{S}^*} D_{j^*}$. Since $D > 4\, c(\mathcal{S}^*)$, at least one $j^* \in \mathcal{S}^*$ satisfies

$$D_{j^*} \;>\; 4\, \frac{c(\mathcal{S}^*)}{|\mathcal{S}^*|} \;=\; 4\, \frac{c(\mathcal{S}^*)}{K}, \tag{23}$$

because $|\mathcal{S}^*| = K$. Denote this center as $j_{\text{large}}^*$ and its cluster $C^* = C(j_{\text{large}}^*)$.

**Step 5: Swapping $j^*$ into $\widehat{\mathcal{S}}$.** Consider the swap

$$\widehat{\mathcal{S}}_{\text{swap}} \;=\; \left( \widehat{\mathcal{S}} \setminus \{ j_{\text{out}} \} \right) \cup \{ j_{\text{large}}^* \}$$

where $j_{\text{out}}$ is whichever center in $\widehat{\mathcal{S}}$ we choose to remove. We must show that for an appropriate choice of $j_{\text{out}}$, the cost $c(\widehat{\mathcal{S}}_{\text{swap}})$ is at least $(r_i - r_i^*)_+$ smaller on average for the points in $C^*$, forcing a net cost reduction large enough to offset any potential cost increase for points outside $C^*$.

In detail, partition $\widehat{\mathcal{S}}$ into $K$ clusters under *Voronoi* assignment:

$$\widehat{C}(j) \;=\; \{ i : j = \arg\min_{x \in \widehat{\mathcal{S}}} d(i, x) \}, \quad j \in \widehat{\mathcal{S}}.$$

Since $|\widehat{\mathcal{S}}| = K$, there must exist at least one $j_{\text{out}} \in \widehat{\mathcal{S}}$ whose cluster $\widehat{C}(j_{\text{out}})$ has weight $\sum\limits_{i \in \widehat{C}(j_{\text{out}})} w_i \;\leq\; \dfrac{1}{K} \sum\limits_{i=1}^{N} w_i$. We remove that $j_{\text{out}}$ and add $j_{\text{large}}^*$.

**Step 6: Net Cost Change Analysis.** After the swap, the net change in cost is $\Delta = \Delta_{in} + \Delta_{out}$. The "in-gain" for points $i \in C^* = C(j^*_{large})$ is bounded by:

$$\Delta_{\text{in}} = \sum_{i \in C^*} w_i \big( d(i, \widehat{\mathcal{S}}_{\text{swap}}) - d(i, \widehat{\mathcal{S}}) \big) \leq - \sum_{i \in C^*} w_i \, (r_i - r_i^*)_+ = -D_{j^*_{large}}. \tag{24}$$

The "out-loss," $\Delta_{out}$, represents the potential cost increase for points not in $C^*$, primarily those that were served by the removed center $j_{out}$. Bounding this term is the most complex part of the proof.

*Remark* D.3 (Bounding $\Delta_{out}$). The analysis in Arya et al. (2001) uses a series of clever applications of the triangle inequality to show that the cost increase, $\Delta_{out}$, is bounded relative to the cost of the optimal solution. A simplified (though non-trivial) result of this bounding shows that for an appropriately chosen $j_{out}$, the increase can be bounded such that:

$$\Delta_{out} \leq \frac{c(\mathcal{S}^*)}{K}. \tag{25}$$

This bound is sufficient to complete the proof. We defer the detailed derivation of this specific bound to the original literature and proceed with this result.

**Step 7: Arriving at a contradiction.** Combining our bounds, the total change in cost is:

$$c(\widehat{\mathcal{S}}_{\text{swap}}) - c(\widehat{\mathcal{S}}) = \Delta_{in} + \Delta_{out} \leq -D_{j^*_{large}} + \frac{c(\mathcal{S}^*)}{K}.$$

From Step 4 (Eq. 23), we know $D_{j^*_{large}} > 4 \frac{c(\mathcal{S}^*)}{K}$. Substituting this in gives:

$$c(\widehat{\mathcal{S}}_{\text{swap}}) - c(\widehat{\mathcal{S}}) < -4 \frac{c(\mathcal{S}^*)}{K} + \frac{c(\mathcal{S}^*)}{K} = -3 \frac{c(\mathcal{S}^*)}{K} < 0.$$

This shows a strict decrease in cost, which contradicts the local optimality of $\widehat{\mathcal{S}}$. Therefore, our initial assumption must be false, and we conclude that $c(\widehat{\mathcal{S}}) \leq 5 \, c(\mathcal{S}^*)$.

**Time Complexity.** At each iteration we try all $O(K\,N)$ possible 1-swaps. By maintaining for each point $i$ its distance to the nearest center in $\mathcal{S}$, we can update the total cost in $O(N)$ time per swap check; hence each pass costs $O(K\,N^2)$. Moreover, letting

$$W_{\text{tot}} = \sum_{i=1}^{N} w_i, \qquad D_{\text{max}} = \max_{i,j} d(i,j),$$

we have

$$0 \leq c(\mathcal{S}) \leq W_{\text{tot}}\, D_{\text{max}}.$$

Since all weights and distances come from the finite input, there is a minimum positive gap $\delta > 0$ between any two distinct cost values. Therefore each improving swap decreases $c(\mathcal{S})$ by at least $\delta$, so there can be at most

$$\frac{W_{\text{tot}}\, D_{\text{max}}}{\delta}$$

such swaps. Altogether the algorithm performs

$$O\big(K\,N^2\big) \times O\big(\tfrac{W_{\text{tot}}\, D_{\text{max}}}{\delta}\big) = \text{poly(input size)}$$

total work, i.e. it runs in polynomial time.

□

*Remark* D.4 (Improved Constants). A more intricate analysis can tighten the factor 5 in Theorem D.1 to 3 or 4. See, e.g., (Gupta & Tangwongsan, 2008; Arya et al., 2001) for classical refinements. The simpler argument here suffices to establish the main principles.

# E. Theoretical Guarantee for AMPO-Coreset

This appendix provides the theoretical motivation for the AMPO-CORESET selection strategy. We first introduce the concept of a coreset and then present a formal theorem showing that, under certain clustering assumptions, this strategy yields a policy with a guaranteed additive bound on its expected reward.

**Coresets for Representative Selection.** The term *coreset* originates in computational geometry and machine learning, referring to a small, weighted subset of data that approximates the entire dataset with respect to a particular objective or loss function (Bachem et al., 2017; Feldman et al., 2020). In the context of AMPO-CORESET, the $K$-means clustering subroutine identifies representative embedding-space regions. By choosing a single worst-rated example from each region, we mimic a coreset-based selection principle: our selected negatives approximate the distributional diversity of the entire batch of responses. This ensures the model receives penalizing signals for all major modes of undesired behavior, mitigating the risk of ignoring infrequent but problematic minority clusters.

## E.1. Additive Guarantee under Bounded-Diameter Clustering

Recall from Appendix C that we use normalized weights

$$W = \sum_{j=1}^{N}(r_{\max} - r_j), \qquad w_i = \frac{r_{\max} - r_i}{W}, \quad \text{so that} \quad \sum_i w_i = 1.$$

This allows Lemma C.1 (Coverage-cost controls negative reward) to give the tight bound $\max_{p \in \mathcal{F}(S)} \sum_i r_i p_i = r_{\max} - L \operatorname{Cost}(\mathcal{S})$. We now show that under the clustering assumption of AMPO-Coreset, the cost term is bounded by $d_{\max}$.

**Theorem E.1** (Additive $Ld_{\max}$-Guarantee for Coreset Selection). *Suppose the $N$ candidate responses can be partitioned into $K$ clusters $\{C_1, \ldots, C_K\}$ in embedding space, each of diameter at most $d_{\max}$:*

$$\max_{i,i' \in C_j} \|\mathbf{e}_i - \mathbf{e}_{i'}\| \leq d_{\max} \quad (j = 1, \ldots, K).$$

*Let the negative set $\mathcal{S}$ be formed by picking one arbitrary index $i_j^- \in C_j$ from each cluster $C_j$. Then, the maximum expected reward achievable by a Lipschitz-compliant policy using this negative set $\mathcal{S}$ is bounded by:*

$$\max_{\substack{p_j = 0 \; (\forall j \in \mathcal{S}) \\ p_i \leq L \min_{l \in \mathcal{S}} \|\mathbf{e}_i - \mathbf{e}_l\|}} \sum_{i=1}^{N} r_i \, p_i \geq r_{\max} - L \, d_{\max},$$

*where $r_{\max} = \max_i r_i$. This guarantees the expected reward is within an additive error of $Ld_{\max}$ of the highest possible reward.*

*Proof.* We use the result from Lemma C.1, which states that the maximum expected reward is $r_{\max} - L \cdot \operatorname{Cost}(\mathcal{S})$, where the cost function uses normalized weights $w_i = (r_{\max} - r_i)/W$. We need to bound $\operatorname{Cost}(\mathcal{S})$ for our chosen $\mathcal{S}$.

$$\operatorname{Cost}(\mathcal{S}) = \sum_{i=1}^{N} w_i \min_{l \in \mathcal{S}} \|\mathbf{e}_i - \mathbf{e}_l\|.$$

For any point $y_i$, it belongs to some cluster $C_j$. By construction, the set of negatives $\mathcal{S}$ contains the point $i_j^-$ from that same cluster. Therefore, the distance from $y_i$ to its closest negative in $\mathcal{S}$ is at most its distance to $y_{i_j^-}$, which is bounded by the cluster diameter:

$$\min_{l \in \mathcal{S}} \|\mathbf{e}_i - \mathbf{e}_l\| \leq \|\mathbf{e}_i - \mathbf{e}_{i_j^-}\| \leq d_{\max}, \quad \forall \, i \in C_j.$$

Since this holds for all points $i$, we can bound the cost:

$$\operatorname{Cost}(\mathcal{S}) = \sum_{i=1}^{N} w_i \underbrace{\min_{l \in \mathcal{S}} \|\mathbf{e}_i - \mathbf{e}_l\|}_{\leq d_{\max}} \leq \sum_{i=1}^{N} w_i \, d_{\max} = d_{\max} \sum_{i=1}^{N} w_i = d_{\max},$$

where the final step uses the fact that the normalized weights sum to 1. Substituting this bound back into the expression for the maximum expected reward gives:

$$\max \sum_{i=1}^{N} r_i \, p_i = r_{\max} - L \operatorname{Cost}(\mathcal{S}) \; \geq \; r_{\max} \; - \; L \, d_{\max},$$

which completes the proof. $\qquad\qquad\qquad\qquad\qquad\qquad\qquad\qquad\qquad\qquad\qquad\qquad\qquad\qquad$ $\square$

**Remark (Distribution-Dependent Guarantee).** The above theorem provides a deterministic guarantee for a fixed set of $N$ points. In practice, we learn from a finite sample of responses drawn from an unknown underlying distribution $\mathcal{D}$. If we learn $K$ clusters from a sufficiently large i.i.d. sample of responses, standard uniform convergence arguments (see, e.g., (Bachem et al., 2017)) show that these empirical clusters will, with high probability, also cover new responses drawn from $\mathcal{D}$. Consequently, for a high fraction of new queries, the policy derived from the coreset selection strategy is expected to achieve a near-optimal reward, with an additive error similar to the $L d_{\max}$ bound.

## F. Optimal Selection Code

In this section we provide the actual code used to compute the optimal selection.

```python
import numpy as np
from scipy.spatial.distance import cdist

def solve_local_search_min_dist_normalized(
    vectors: np.ndarray,
    rating: np.ndarray,
    k: int,
    max_iter: int = 100,
    random_seed: int = 42
):
    # Normalize ratings
    rating_min = np.min(rating)
    rating_max = np.max(rating)
    rating_normalized = (rating - rating_min) / (rating_max - rating_min) if rating_max >
        rating_min else np.zeros_like(rating) + 0.5

    # Identify top-rated point
    excluded_top_index = int(np.argmax(rating_normalized))

    # Reduce dataset
    new_to_old = [idx for idx in range(len(rating_normalized)) if idx !=
        excluded_top_index]
    vectors_reduced = np.delete(vectors, excluded_top_index, axis=0)
    rating_reduced = np.delete(rating_normalized, excluded_top_index)

    # Compute L2 distances and normalize
    if len(rating_reduced) == 0:
        return excluded_top_index, None, [], [], []
    distance_matrix = cdist(vectors_reduced, vectors_reduced, metric='euclidean')
    distance_matrix /= distance_matrix.max() if distance_matrix.max() > 1e-12 else 1

    # Compute weights
    mean_rating_reduced = np.mean(rating_reduced)
    w = np.exp(mean_rating_reduced - rating_reduced)

    # Local search setup
    def compute_objective(chosen_set):
        return sum(w[i] * min(distance_matrix[i, j] for j in chosen_set) for i in range(
            len(w)))

    rng = np.random.default_rng(random_seed)
    all_indices = np.arange(len(rating_reduced))
```

```python
current_set = set(rng.choice(all_indices, size=k, replace=False)) if k < len(
    rating_reduced) else set(all_indices)
current_cost = compute_objective(current_set)

# Local search loop
improved = True
while improved:
    improved = False
    best_swap = (None, None, 0)
    for j_out in list(current_set):
        for j_in in all_indices:
            if j_in not in current_set:
                candidate_set = (current_set - {j_out}) | {j_in}
                improvement = current_cost - compute_objective(candidate_set)
                if improvement > best_swap[2]:
                    best_swap = (j_out, j_in, improvement)
    if best_swap[2] > 1e-12:
        current_set.remove(best_swap[0])
        current_set.add(best_swap[1])
        current_cost -= best_swap[2]
        improved = True

chosen_indices_original = [new_to_old[j] for j in sorted(current_set)]
rejected_indices_original = [new_to_old[j] for j in sorted(set(all_indices) -
    current_set)]
return excluded_top_index, chosen_indices_original[0], rejected_indices_original[:k],
    chosen_indices_original, rejected_indices_original
```

# G. Visualization of t-SNE embeddings for Diverse Responses Across Queries

In this section, we showcase the performance of our method through plots of TSNE across various examples. These illustrative figures show how our baseline Bottom-k Algorithm (Section 5.1) chooses similar responses that are often close to each other. Hence the model misses out on feedback relating to other parts of the answer space that it often explores. Contrastingly, we often notice diversity of response selection for both the AMPO-OPTSELECT and AMPO-CORESET algorithms.

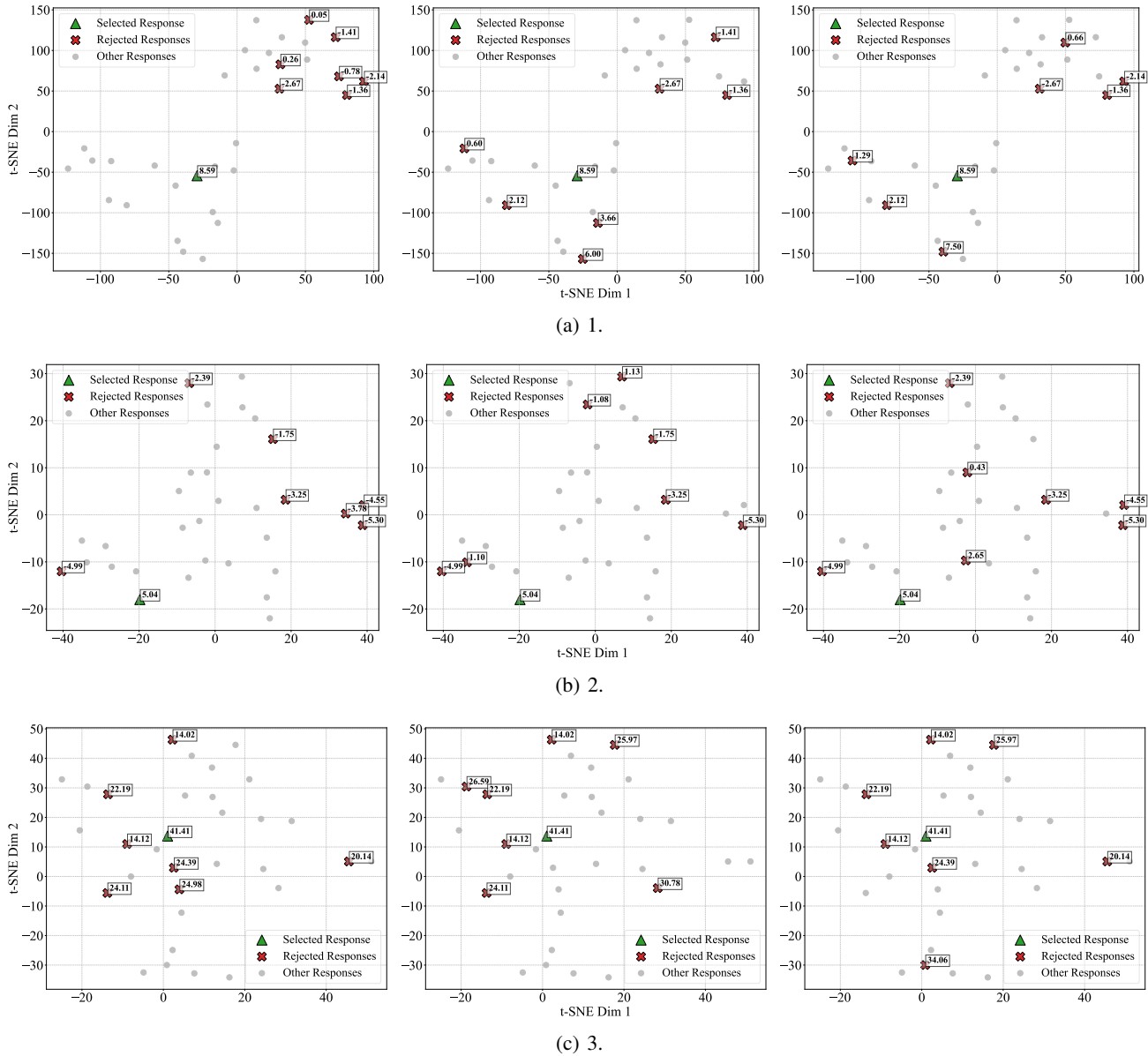

*Figure 6.* t-SNE visualization of projected high-dimensional response embeddings into a 2D space, illustrating the separation of actively selected responses. (a) AMPO-BottomK (baseline). (b) AMPO-Coreset (ours). (c) Opt-Select (ours). Traditional baselines select many responses close to each other based on their rating, providing insufficient feedback to the LLM during preference optimization. In contrast, our methods optimize for objectives including coverage, generation probability, and preference rating.

