# OpenReview forum: "AMPO: Active Multi Preference Optimization for Self-play Preference Selection"
_ICML.cc/2025/Conference — ICML 2025 poster_

### Official Review · Reviewer_P2gm · 2025-03-03

**Overall Recommendation:** 2

**Summary:**

The paper introduces AMPO, a method for aligning LLM using multi-response preference selection instead of traditional pairwise comparisons. It combines on-policy data generation, a group-contrastive loss, and active subset selection to train models. Various selection strategies, called bottom-k, coreset clustering, and Opt-Select, was proposed to improve preference learning by focusing on diverse and extreme responses. Experiments on Llama 8B show improved performance on AlpacaEval and other benchmarks.

**Claims And Evidence:**

Yes

**Essential References Not Discussed:**

Since the paper focuses on On-Policy Self-Play, I believe more related works in this area should be discussed. Currently, the related work section includes only four references, none of which are recent. This makes it less clear how the paper contributes to the field.

**Experimental Designs Or Analyses:**

Yes.

**Methods And Evaluation Criteria:**

Yes

**Other Comments Or Suggestions:**

The dataset link in the abstract leads to 404.

**Other Strengths And Weaknesses:**

Strengths:

- The paper is well-written and easy to understand.
- The proposed method achieves state-of-the-art results on AlpacaEval.

Weaknesses:

- Perhaps the biggest concern is that the method is limited to scenarios where a reward model is available to evaluate and score all answers. This contradicts the paper's claim that the proposed solution is superior to pairwise preference. Pairwise preference addresses challenges in collecting real human feedback for RLHF.

**Questions For Authors:**

How does the choice of the reward model affect performance? Could the observed improvement be primarily due to a well-designed reward model rather than the proposed method itself? To better isolate the contribution of the method, it would be beneficial to design an experiment that compares its performance across different reward models.

**Relation To Broader Scientific Literature:**

The key contribution is a unified pipeline for self-play alignment considering multi-preference. It could inspire more discussion in the raising topic of on-policy self-play alignment.

**Theoretical Claims:**

No due to time limit.

---

> ### Author Rebuttal · Authors · 2025-04-01
>
> We thank Reviewer P2gm for their insightful and helpful feedback. We appreciate the opportunity to address the concerns raised. We have prepared a detailed point-by-point response with supporting experiments in the full rebuttal document (provided separately) and offer a summary here. We hope these clarifications adequately address the reviewer's points and would be grateful if the reviewer would consider increasing their score should they find the responses satisfactory.
>
> **Response to Weakness 1: Method Limitation to Reward Model Scenarios vs. Pairwise Preference**
>
> We understand the concern regarding the reliance on a reward model versus traditional pairwise human feedback. Our detailed response (Part 1 of the rebuttal document) clarifies this in three ways:
>
> 1.  **Alignment with Modern Practices:** We highlight that contemporary LLM alignment commonly uses an initial phase of pairwise preference collection (human or LLM) to train a reward model (RM). This trained RM is then used to generate scalar scores for many candidate responses scalably and cost-effectively. This approach, employed by methods like SimPO [Meng et al., 2024], RPO [Adler et al., 2024], SPPO [Wu et al., 2024], and RSPO [Tang et al., 2025], *extends* rather than contradicts pairwise methods. Our method, AMPO, builds upon this standard practice.
> 2.  **Validity of LLM Feedback:** We reference literature (e.g., "LLM as Judge" [Zheng et al., 2023], Prometheus [Kim et al., 2023], RewardBench [Lambert et al., 2024]) demonstrating that scores from well-trained RMs or strong LLM judges correlate highly with human preferences, serving as effective proxies.
> 3.  **Empirical Robustness (Detailed in Document):**
>     *   **Changing the Reward Model (Part 2 & Table 1):** We show empirically that AMPO maintains strong performance even when switching from an 8B Skywork RM to a 2B GRM-based RM [Yang et al., 2024], indicating robustness to the specific RM used.
>     *   **Removing Rating Weight (Part 3 & Table 2):** We demonstrate through ablation (using L0 weighting) that using reward scores *only* to classify responses into "select" and "reject" pools achieves performance nearly as strong as using the precise scores (L1 weighting). This suggests the primary value comes from the *preference signal* (select vs. reject), aligning with findings that LLMs excel as classifiers [Farebrother et al., 2024], rather than relying solely on perfectly calibrated scores.
>
> In summary, AMPO leverages the standard and scalable practice of using reward models (often derived from initial pairwise data) and is robust to both the specific reward model and the precise granularity of the scores.
>
> **Response to Weakness 2: Broken Dataset Link**
>
> Thank you for pointing this out. We sincerely apologize for the typo in the abstract's link. The links in the contributions section were correct. The correct URLs are:
>
> *   **OPT Select:** https://huggingface.co/datasets/Multi-preference-Optimization/AMPO-OPT-Selection
> *   **Coreset:** https://huggingface.co/datasets/Multi-preference-Optimization/AMPO-Coreset-selection
>
> We will correct this in the revised manuscript.
>
> **Response to Weakness 3: Isolating Method Contribution vs. Reward Model Choice**
>
> We appreciate this insightful question about isolating AMPO's contribution. As summarized above and detailed in our full rebuttal (Part 2 & Table 1):
>
> 1.  **Cross-RM Experiments:** We conducted experiments using two distinct reward models: the 8B Skywork RM and a 2B GRM-based RM [Yang et al., 2024]. AMPO demonstrated strong and comparable performance across both, particularly on AlpacaEval 2. This suggests the observed improvements stem significantly from the AMPO selection and optimization framework itself, not just the quality of one specific reward model.
> 2.  **AMPO's Use of RM:** Furthermore, AMPO uses the reward model primarily to establish *relative rankings* – identifying the best response and selecting a diverse set of lower-quality responses for contrast. The optimization then maximizes the likelihood of the best while minimizing the joint likelihood of the negative set. This reliance on relative ranking, rather than direct supervision by noisy scalar values, enhances stability and generalizability.
>
> The full experimental results comparing both reward models are included in the rebuttal document and will be in the updated manuscript.
>
> ---
>
> We hope these summaries, elaborated upon in the full rebuttal document, effectively address the reviewer's concerns. We believe AMPO offers a valuable contribution by efficiently leveraging multi-preference data within modern alignment paradigms. We respectfully request the reviewer consider these responses in their final evaluation.
>
> We provide a detailed document outlining our response here: <https://drive.google.com/file/d/1NzYto9Nl7XyIDIMJKtk3uD2H_yHG2hun/view?usp=sharing>.

---

### Official Review · Reviewer_gkZq · 2025-03-13

**Overall Recommendation:** 2

**Summary:**

The paper introduces the AMPO (Active Multi-Preference Optimization) method, which aims to improve the alignment performance of large language models (LLMs) through active negative sample selection in multi-preference optimization. The main contributions are: proposing several active selection strategies to choose representative response subsets for more efficient multi-preference alignment training; demonstrating that coverage-based negative selection can systematically suppress low-reward modes and maximize expected reward. The paper achieves advanced results on multiple benchmarks and releases relevant datasets for future research.

**Claims And Evidence:**

AMPO's active selection strategies (such as AMPO-Coreset and AMPO-OptSelect) can cover a wider semantic space and avoid selecting overly similar responses, thus providing richer training signals. This is intuitively demonstrated through t-SNE visualization.

**Essential References Not Discussed:**

N/A

**Experimental Designs Or Analyses:**

The paper validates AMPO's effectiveness through benchmarks and compares it with existing methods (SIMPO). However, it only trains and tests on one benchmark model, and the baseline for comparison includes only one method, making the comparison less convincing.

**Methods And Evaluation Criteria:**

The paper selected three benchmarks (AlpacaEval, Arena-Hard, and MT-Bench) to evaluate model performance, but it may lack assessment on datasets targeting mathematics, code, and reasoning.

**Other Comments Or Suggestions:**

Please refer to Other Strengths And Weaknesses

**Other Strengths And Weaknesses:**

Strengths: The paper proposes multiple different selection strategies for negative samples in preference optimization to improve training efficiency. It proves that coverage of subsets in semantic space is an important consideration for negative sample selection.

Weaknesses:
1. The number of experiments is small, lacking persuasiveness in a wider range of scenarios.
2. There is no explanation of the size of the training data.
3. The selection strategy based on optimization problems (Opt-Select) is not better than the strategy based on clustering results (AMPO-Coreset), which is clearly more understandable in theory and simpler in computation. The paper fails to explain the advantages of the Opt-Select method over the AMPO-Coreset method.

**Questions For Authors:**

1. Are there other multi-preference optimization related works that can serve as experimental baselines?
2. What are the advantages of Opt-Select compared to AMPO-Coreset?

**Relation To Broader Scientific Literature:**

AMPO continues the research direction of multi-preference optimization, providing richer training signals by considering multiple "good" and "bad" responses simultaneously, which resonates with recent related work (such as SWEPO).

**Theoretical Claims:**

The paper proves that the AMPO-OptSelect strategy can maximize the model's expected reward. For large-scale problems, it proposes a local search algorithm that can approximate the optimal solution with polynomial time complexity, a conclusion supported by theory.

---

> ### Author Rebuttal · Authors · 2025-03-31
>
> Thank you for your insightful and helpful review. We have conducted several additional experiments, detailed in the full rebuttal document (provided separately), and summarize the key findings here. We hope these responses adequately address your points and would be grateful if you would consider increasing your score.
>
> **Response to Weakness 1: Limited Number of Experiments**
>
> To broaden the validation of AMPO, we have significantly expanded our experiments as detailed in the full rebuttal document:
>
> 1.  **New Model Setting:** We added results for **Mistral-7B-Instruct-v0.2**, demonstrating AMPO's effectiveness beyond Llama-3 (see Table 1 in the detailed document).
> 2.  **Reward Model Ablations:** We tested AMPO with a **2B GRM-Reward-LM** [Yang et al., 2024] alongside the original 8B Skywork RM, showing robustness to the RM choice (see Table 2 in the detailed document).
> 3.  **Single vs. Multiple Positive Responses:** Experiments confirm that contrasting **1 positive vs. k negatives** consistently outperforms k-vs-k settings, validating our primary approach (see Tables 3 & 4 in the detailed document).
> 4.  **Variation with k (Number of Negatives):** We analyzed performance as $k$ increases in the 1-vs-k setup. Results show diminishing returns after $k=3$, supporting our claim that selecting a subset (like 1 vs 7) is efficient (see Table 5 in the detailed document).
> 5.  **Variation with/without Rating Weight:** Ablations comparing $\ell_1$ (using scores) vs. $\ell_0$ (classification only) weighting show marginal differences, indicating the main benefit comes from the select/reject preference signal (see Table 6 in the detailed document).
>
> As a sample of the results obtained, we provide our results in the new model setting as below:
>
> | Method              | Reward Model      | AlpacaEval 2 LC (%) | AlpacaEval 2 WR (%) | Arena-Hard WR (%) | MT-Bench GPT-4 |
> | :------------------ | :---------------- | :-----------------: | :-----------------: | :---------------: | :-------------: |
> | Base                | -                 | 17.1                | 14.7                | 12.6              | 7.5             |
> | Best vs Worst (SimPO)| Skywork-Reward-LM | 30.1                | 32.3                | 21.1              | 7.56            |
> | AMPO-Bottomk        | Skywork-Reward-LM | 32.1                | 37.0                | 22.1              | 7.7             |
> | AMPO-Coreset        | Skywork-Reward-LM | 32.8                | 37.3                | 22.6              | **7.75**        |
> | AMPO-Opt-Select     | Skywork-Reward-LM | **33.1**            | **37.8**            | **22.8**          | 7.72            |
>
> *Table 1: Performance on Mistral-7B-Instruct. Best results in **bold**.*
>
>
>
> We believe these comprehensive experiments substantially strengthen the paper's empirical validation across different scenarios.
>
> **Response to Weakness 2: Explanation of Training Data Size**
>
> We apologize for the lack of clarity. We use the standard **UltraFeedback dataset**.
> *   The initial dataset has 61.1k training examples. Due to filtering for AMPO strategies requiring $k=7$ rejected responses, our final dataset consists of **59.2k training** and **1.92k validation** examples.
> *   Using UltraFeedback is standard practice (e.g., SimPO [Meng et al., 2024], RPO [Adler et al., 2024]).
> *   The specific filtered datasets are released (links provided in the detailed document).
>
> **Response to Weakness 3 & Question 2: Advantages of Opt-Select over Coreset**
>
> While AMPO-Coreset is simpler and strong at T=0, AMPO-Opt-Select offers key advantages:
>
> 1.  **Stability at Higher Temperatures:** Opt-Select consistently outperforms Coreset at typical sampling temperatures (T=0.25-0.75), indicating greater robustness in practical use cases (see Figure 1 in the detailed document).
> 2.  **Robustness Across Reward Models:** Opt-Select showed significantly better performance relative to Coreset when using the smaller GRM-Reward-LM, suggesting it's less sensitive to the specific RM (see Table 2 in the detailed document).
>
> Therefore, Opt-Select provides more robust alignment, particularly under varied generation settings and reward models. We will clarify this trade-off in the revision.
>
> **Response to Question 1: Other Multi-Preference Baselines**
>
> Our experiments include **Best-vs-Worst (SimPO)** adapted to multi-response data and **AMPO-Bottomk**. While other methods exist, many recent ones like **RPO-bwd** are equivalent to **InfoNCA**, which uses a different loss structure (multi-class CE) than AMPO's group contrastive approach.
>
> ---
>
> We hope these clarifications and the additional experiments detailed in the full rebuttal document address your concerns. We believe AMPO offers a valuable contribution to multi-preference alignment and respectfully request you consider these points in your final assessment.
>
> We provide a document outlining our experimental results here: <https://drive.google.com/file/d/1XV9v5dZAQfO9mggCuW85mu_0wgoB6gdF/view?usp=sharing>

---

### Official Review · Reviewer_kqJE · 2025-03-18

**Overall Recommendation:** 3

**Summary:**

This paper studies the multi-preference optimization problem in which two sets of helpful and undesired responses are contrasted during self-play alignment. The authors propose Active Multi-Preference Optimization (AMPO), a framework comprising on-policy generation, a multi-preference group-contrastive loss, and active subset selection. AMPO scores and embeds large pools of candidate responses, then selects a small but informative subset covering reward extremes and distinct semantic clusters for preference optimization. The authors provide theoretical guarantees and empirical results showing AMPO achieves good performance on benchmarks (MT-Bench, AlpacaEval2, and Arena-Hard v0.1) with Llama 8B while  GPT-4-Preview-1106 as a judge model to give scores for responses.

**Claims And Evidence:**

The claims made in the paper are supported by clear and convincing evidence, including both theoretical and empirical results.

**Essential References Not Discussed:**

The key related works are discussed.

**Experimental Designs Or Analyses:**

I do not find any issue with the method and evaluation criteria.

**Methods And Evaluation Criteria:**

I do not find any issue with the method and evaluation criteria, except that the authors should have tested their methods across different models instead of only Llama 8B.

**Other Comments Or Suggestions:**

Check the weaknesses raised in **Other Strengths And Weaknesses**.

**Minor comment or suggestions:**
1. For completeness, the authors should briefly add the most relevant related work in the main paper.
2. The first two algorithms use $N$ to denote the number of responses, whereas the last two use $n$.

**Other Strengths And Weaknesses:**

#### **Strengths of paper:**
1. This paper considers the multi-preference optimization problem, whose efficient solutions will be useful for LLM alignment.

2. The authors propose an AMPO framework that combines on-policy generation, multi-preference optimization, and active learning for efficient multi-preference optimization. They overall propose four AMPO algorithms: AMPO-BottomK, AMPO-Coreset, AMPO-OptSelect (MIP), and AMPO-OptSelect (Coordinate Descent) ) in which the last two algorithms optimize expected reward for active subset selection. The authors also provide theoretical guarantees under some assumptions of optimality for the proposed algorithms

3. The authors also verify the performance of the proposed algorithms via experiments using different benchmarks and release their datasets on Hugging Face.


#### **Weaknesses of paper:**
1. The motivating examples of why one should consider group-based preference optimization for robust alignment when scores for each response are available as scores have more information and one can directly use them than only considering group-based preference. Further, the scores may not be accurate or even have an inherent bias from the rating model used in the AMPO framework.

2. It is unclear why only one positive response and $k$ negative responses are selected in each of the algorithms; why not use the range of scores (very high, high, low, and very low) and choose them across the response space? There should be some theoretical reason or ablations to justify the choice made in the paper.

3. The paper's title is a bit misleading as it gives the impression that the paper considers the problems when only human preference feedback is available (i.e., binary feedback telling which one is better between two responses).

4. There is a mismatch in the definition of $S^{-}$ in Line 135 and Eq. (5).

5. **Limited experiments:** Only Llama 8B is used for the experiment. Additional experiments should be added to show how hyperparameters of the proposed algorithms (N and k) influence overall performance.

6. **About assumptions:** It is unclear how well the assumptions taken in the paper hold in practice.

**Questions For Authors:**

Please address the weaknesses raised in **Other Strengths And Weaknesses**.

**Relation To Broader Scientific Literature:**

To the best of my knowledge, the paper covers the relevant related scientific literature.

**Theoretical Claims:**

I have made a bird's-eye pass of the proofs of the theoretical results in the Appendix, and they appear correct to me. However, I have not read the proofs in detail.

---

> ### Author Rebuttal · Authors · 2025-03-31
>
> Thank you for your insightful and helpful review, and for acknowledging the strengths of our work. We appreciate the opportunity to address your concerns point-by-point. We have prepared a detailed response with additional experiments in the full rebuttal document (provided separately) and summarize the key clarifications below. We hope these responses are satisfactory and would be grateful if you would consider increasing your score.
>
> **Response to Weakness 1: Group Preference vs. Scores & Reward Model Bias**
>
> This is an insightful point. Our investigation, detailed in the full rebuttal document, shows:
>
> 1.  **Score Usefulness:** Using exact reward scores ($\ell_1$-weighting) provides only a marginal gain over using scores simply to classify responses ($\ell_0$-weighting), primarily on AlpacaEval 2 (see Table 1 in the detailed document). This suggests the main value comes from identifying the preferred group, aligning with literature viewing RMs as effective classifiers rather than precise regressors [Farebrother et al., 2024].
> 2.  **Reward Model Bias:** We tested AMPO with two different reward models (8B Skywork and 2B GRM). The results demonstrate robustness, with AMPO performing well with both, indicating low sensitivity to the specific RM choice or its potential biases (see Table 2 in the detailed document).
>
> **Response to Weakness 2: Selection of Single Positive Response**
>
> Our choice to select the single best positive response against multiple negatives is motivated by:
>
> 1.  **Theoretical Optimality:** Our analysis (Section B.1 in the paper) shows that maximizing expected reward involves concentrating probability mass on the single best-rated response while suppressing negative ones.
> 2.  **Empirical Validation:** Ablations comparing 1-vs-k settings against k-vs-k settings (e.g., 1v7 vs 4v4, 1v3 vs 2v2) consistently show that the single positive approach yields better performance across benchmarks (see Tables 3 & 4 in the detailed document).
>
> Both theory and experiments support focusing on the single highest-reward response for optimization.
>
> **Response to Weakness 3: Misleading Title (Human vs. LLM Feedback)**
>
> We acknowledge the historical context. However, the current LLM alignment field widely uses LLM-generated preferences or reward models trained on initial (often human) pairwise data as a standard, scalable practice (e.g., SimPO [Meng et al., 2024], RPO [Adler et al., 2024], SPPO [Wu et al., 2024]). Furthermore, strong correlation between LLM judges and human preferences is well-documented [Zheng et al., 2023; Kim et al., 2023; Lambert et al., 2024]. Given this contemporary context, we believe the title accurately reflects our contribution within the standard paradigm where LLM-derived feedback is common and validated.
>
> **Response to Weakness 4: Typo in Line 135 / Eq. (5)**
>
> Thank you for catching this. We acknowledge the typo and will correct it in the final version to match Equation 5.
>
> **Response to Weakness 5: Limited Experiments (Model & Hyperparameters)**
>
> We have addressed this by:
>
> 1.  **Adding Mistral-7B Experiments:** We now include results for Mistral-7B-Instruct-v0.2, demonstrating AMPO's effectiveness on another widely used model (see Table 5 in the detailed document).
> 2.  **Analyzing Hyperparameter k:** We provide experiments varying the number of negative samples ($k$) for AMPO-Opt-Select (1vs3, 1vs5, 1vs7). Results show consistent improvement with $k$, but diminishing returns after $k=3$, supporting the efficiency of our subset selection (see Table 6 in the detailed document).
>
> **Response to Weakness 6: Practicality of Assumptions**
>
> We appreciate the concern regarding our theoretical assumptions. As detailed in the full rebuttal document:
> *   Assumptions like (A1) Lipschitz, (A3) Finite Support, and (A5) Bounded-Diameter Clusters are standard in related theoretical analyses (e.g., metric bandits, clustering, policy learning) to enable formal guarantees.
> *   We acknowledge the gap between these idealized assumptions and real-world LLM behavior. Our method is empirically robust even when these assumptions hold only approximately.
> *   We will explicitly discuss these limitations and the practical implications in the revised manuscript to clarify the applicability.
>
> **Response to Minor Comments:**
>
> *   We will ensure consistent notation (N vs k) in the final version.
> *   We will review the main paper to ensure key related work is adequately summarized.
>
> ---
>
> We hope these clarifications and the additional experiments detailed in the full rebuttal document effectively address your concerns. We believe AMPO offers a novel and empirically strong approach to multi-preference optimization. We respectfully request you consider these points in your final assessment, and increase your score should you find that we have addressed your concerns.
>
> Please find attached a document where we detail our response: <https://drive.google.com/file/d/1VQvnqa7dsX8RunB2u71tsOPzye7JkWBe/view?usp=sharing>

---

> > ### Comment · Reviewer_kqJE · 2025-04-09
> >
> > Thank you for your detailed response. Please include these discussions in the revised version of your paper. Since my concerns have been addressed, I have increased my *Overall Recommendation* to 3.

---

> > > ### Author Response · Authors · 2025-04-09
> > >
> > > Thank you for your thoughtful and constructive feedback, we really appreciate the time and care you put into the review. We're glad the clarifications helped address your concerns, and we're grateful for your updated recommendation. We’ll incorporate the key points from the rebuttal into the revised paper and believe these changes significantly strengthen the work. We hope other reviewers and AC reviewing the paper will also take these updates into account when making their final assessments

---

### Decision · Program_Chairs · 2025-05-01

**Decision:**

Accept (poster)

**Comment:**

Summary
The paper introduces AMPO, a framework that aligns large language models with a set of responses rather than pairwise preference data. For each prompt, it selects the single best response and contrasts it against an actively chosen subset of poorer answers drawn from on‑policy generations. Several active‑selection strategies are analysed, and variants of the framework are evaluated on multiple public benchmarks. While the empirical results are promising, the study relies on a limited set of baselines.

Recommendation
Considering (i) one clear weak‑accept review, (ii) two weak‑reject reviews whose main concerns appear to be addressed in the rebuttal (though the reviewers have not yet updated their scores), and (iii) additional results that support the performance claims, I recommend weak accept pending minor revisions. I highly recommend including more baselines.